psychology/behaviour

description experience gap, probability perception, inference, decision making, risk

**Author for correspondence:**
Thomas Ward Elston
e-mail: telston@nurhopsi.org

# Outcome uncertainty influences probability perception and risk attitudes

Thomas Ward Elston[1,3], Ian Grant Mackenzie[2] and Victor Mittelstädt[2]

[1]Animal Physiology Unit, Institute for Neurobiology, and [2]Department of Biological Psychology, University of Tübingen, Tübingen 72070, Germany
[3]Helen Wills Neuroscience Institute, University of California at Berkeley, Berkeley, CA 94709 USA

 TWE, 0000-0001-9833-5121

Subjective inferences of probability play a critical role in decision-making. How we learn about choice options, through description or experience, influences how we perceive their likelihoods, an effect known as the description–experience (DE) gap. Classically, the DE gap details how low probability described options are perceptually inflated as compared to equiprobable experience ones. However, these studies assessed probability perception relative to a 'sure-bet' option, and it remained unclear whether the DE gap occurs when humans directly trade-off equiprobable description and experience options and whether choice patterns are influenced by the prospects of gain and loss. We addressed these questions through two experiments where humans chose between description and experience options with equal probabilities of either winning or losing points. Contrary to early studies, we found that gain-seeking participants preferred experience options across all probability levels and, by contrast, loss-mitigating participants avoided the experience options across all probability levels, with a maximal effect at 50%. Our results suggest that the experience options were perceived as riskier than descriptive options due to the greater uncertainty associated with their outcomes. We conclude by outlining a novel theory of probabilistic inference where outcome uncertainty modulates probability perception and risk attitudes.

## 1. Introduction

We are regularly confronted with choices that involve balancing risks—for example, whether to get vaccinated given the risk of certain side effects. We can learn how probable such outcomes are either by explicit description (we are told that the probability of side effects is 1%) or inferred via experience (personally experiencing side effects or knowing people who have).

Interestingly, the way we learn about outcome probabilities, through description or experience, influences how we perceive them [1,2]. Specifically, human monetary gambling studies suggest that rare described options are perceived as more likely to occur than they actually are whereas the opposite is the case with options whose probabilistic outcomes have been inferred through repeated experience [1]. This phenomenon, where probability perception depends on whether outcome probabilities were learned either through description or experience, is known as the description–experience (DE) gap [3].

The DE gap has traditionally been studied in the context of monetary gambles where human participants decide between two options on each trial—often between either a described (e.g. 50% chance of winning $100) or experience (e.g. a red square whose probability has to be inferred through repeated sampling) option and a described safe, 'sure-bet' option (e.g. 100% chance of winning $50) (for reviews, see: [2,3]). Participants' subjective beliefs about description/experience option outcome probabilities are typically inferred through the concept of a certainty equivalent (CE) which is defined relative to a 'sure-bet' option. CEs are calculated by finding the description/experience stimulus that is chosen with the same likelihood of the 'sure-bet' option and then used to derive participants' probabilistic perceptions. The classic finding from these experiments has been an interaction of probability and stimulus type: low probability described options are perceived as more likely to occur than low probability experience options and the reverse at high probability: high probability described options are perceived as less likely than high-probability experience options. Although factors such as sampling error and recency bias have been shown to contribute to the DE gap, it appears to persist even when these factors are controlled (for review, see: [2,3]).

One methodological innovation in studying the DE gap is the notion of equiprobable gambles—situations where subjects must choose between a described and experience option that have objectively equal outcome likelihoods [4,5]. Equiprobable gambles are appealing because they allow investigators to isolate the influence of stimulus type (description, experience) on probability perception by directly comparing choice patterns (e.g. $p$(Choose Experience)). This allows investigators to discern whether the processes underlying the DE gap can be observed at the level of raw choice without the intermediate step of computing a CE relative to a safe option. To our knowledge, only two prior studies have used the equiprobable concept to study the DE gap. Critically, as reviewed in the following, their results call into question the orientation and profile of classic accounts of the DE gap (figure 1c for choice patterns predicted by the classic inverse-S profile DE gap in the equiprobable context).

First, Heilbronner & Hayden [4] used the equiprobable by requiring non-human primates (NHPs) to decide between equiprobable description (partially filled bars) and experience (nature pictures) options while seeking gains (water) across range probabilities. Although they report evidence of a stimulus type*probability DE gap profile, Heilbronner and Hayden's NHPs appeared to perceptually inflate the payoff probabilities of experience options as compared to described ones at all probability levels but especially at low probabilities (e.g. 20%). This was opposite to Hertwig et al.'s [1] early and more recent (e.g. [2]) reports that the DE gap is oriented towards described options and instead suggests that the DE gap may be experience-oriented, at least in NHPs. One additional caveat in interpreting Heilbronner & Hayden's [4] results is that they did not directly assess choice patterns within each equiprobable condition—they collapsed across all probability levels and stimulus types, including a 'sure-bet' option, rather than computing the DE gap directly for each probability level (e.g. $p$(Choose Experience) during 20% equiprobable gambles). Thus, it remains unclear whether the DE gap takes on an inverse-S-shaped stimulus type*probability profile and whether the DE gap is oriented towards description or experience options when humans directly compare equiprobable choice patterns across several probability levels.

Second, Ludvig & Spetch [5] assessed how stimulus type influenced the perception of a 50% option in the contexts of mitigating losses (of points) and seeking gains (of points). They found that humans deciding between a safe option and either a risky description or experienced option of equal probability (both options had an objective outcome probability of 50%) strongly preferred the experience stimuli when seeking gains but preferred the described options when mitigating losses. The preference for experience stimuli over probabilistically identical description stimuli (when seeking gains) indicates that participants may perceive experience stimuli as more probable to occur—implying that outcome probabilities of experience stimuli are perceptually inflated. These results were surprising because classic accounts of the DE gap in humans consistently reported a description-oriented effect that was most pronounced at low (less than 30%) probabilities. These results also highlight the possibility that the contexts of seeking gains/mitigating losses may play a major role in human preferences for description and experience options. However, Ludvig & Spetch [5] only assessed choices at the 50% probability level, leaving the question open as to whether the DE gap is

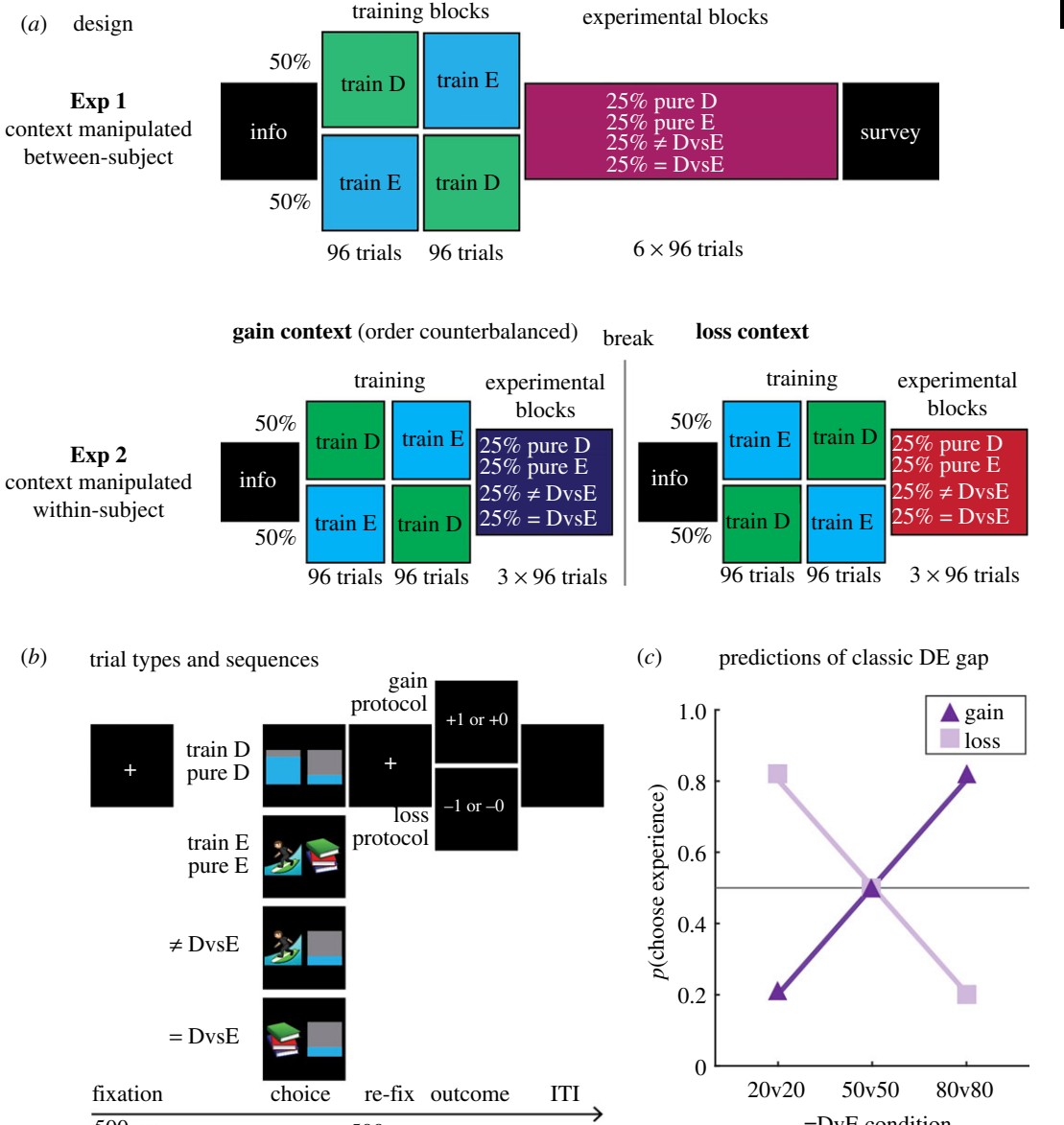

**Figure 1.** Experimental design and hypothesis. (*a*) Session organization of the between subject (Exp 1) and within-subject (Exp 2) manipulations. (*b*) Example of the structure of individual trial types and example stimuli. Sequences were identical between the gain and loss contexts in both experiments. (*c*) Schematic of the choice patterns predicted by the classic DE gap during the equiprobable trials (=DvE in B). The gain context was hypothesized to elicit a bias towards the description stimuli selectively at low probabilities and an aversion to them at high probabilities, consistent with an inverse-S probability weighting function. We hypothesized the mirrored pattern for the loss context. We operationalized the DE gap as a participants absolute choice bias from 50%.

best characterized as a stimulus type*probability interaction where described stimuli are perceptually inflated at low probabilities. In sum, recent studies suggest that the probability distortions underlying the DE gap may affect choice differently when humans directly trade-off description and experience options of equal probability in different risk contexts (seeking gains or mitigating losses).

Motivated by these issues, the major questions of the present study were (i) to assess whether the classic DE gap pattern will present when humans directly decide between equiprobable described and experience options and (ii) to assess whether/how equiprobable choice patterns are influenced by the contexts of seeking gains and mitigating losses (figure 1*c*). To address these questions, we conducted two behavioural experiments where participants either sought gains or mitigated losses by choosing between description and experience options of different probabilities (Exp 1: $N = 60$; pre-registered Exp 2: $N = 60$). To more fairly compare choice patterns between description and experience options, we used non-verbal stimuli for both types (figure 1*a,b* for design and example stimuli). Our critical

experimental condition involved equiprobable gambles—situations where participants chose between described and experience options with objectively equal probabilities. This allowed us to isolate the influence of stimulus type (description, experience) on probability perception by directly comparing choice patterns (e.g. $p$(Choose Experience)) as a function of probability and context (seeking gains or mitigating losses).

Thus, our central analyses concerned choice patterns during 'equal description versus experience' trials. On the basis of classic DE gap studies showing that low probability described options are perceptually inflated as compared to equiprobable experience ones (and the reverse at high probabilities), we predicted that gain-seeking participants would prefer the 'description' bar stimuli during low probability 'equal descriptive versus experience' trials (e.g. trials where the payoff probability of both options is 20%) and, by contrast, would prefer the experience emoji stimuli during high-probability 'equal descriptive versus experience' trials (e.g. trials where the payoff probability of both options is 80%). We predicted the opposite pattern would emerge for participants mitigating losses.

To foreshadow the results, contrary to the classic description-oriented DE gap that is most pronounced at low probabilities, we observed an experience-oriented DE gap that was maximal at 50%. As is elaborated in our General Discussion, these findings suggest that, at least in the equiprobable context, the DE gap is experience-oriented and is not limited to rare (low probability) events.

# 2. Method

## 2.1. Participants

### 2.1.1. Experiment 1

Sixty German-speaking psychology students (50 female; mean age: 21.05, range: 18–34) at the University of Tübingen were recruited via campus advertisements, social media and internal departmental e-mail lists. They received course credits for their participation. Thirty participants were randomly allocated to the gain-oriented context and the remaining 30 were randomly allocated to the loss-oriented context. Each participant was tested in a single experimental session lasting approximately 30 min. No subject participated in both contexts. See figure 1a for an overview of the session organization.

### 2.1.2. Experiment 2

This experiment was conducted after Experiment 1 and pre-registered on the Open Science Framework, the link to which can be found here: https://osf.io/a8zf6/?view_only=b43e278fe15d4fe5a51dd52ee 4bc9c30. Participant recruitment (60 human, 44 female; mean age: 23.52, range: 19–59) was identical to Experiment 1, with the exception that participants had the additional incentive of the opportunity to win 10€ vouchers for their participation. Contrary to the between-subject design used in Experiment 1, all participants experienced both the gain and loss contexts in a split half, counterbalanced order. Thirty participants first went through the gain context followed by the loss context; the remaining 30 participants first went through the loss context followed by the gain context. No participant of Experiment 2 participated in Experiment 1. See figure 1a for an overview of the session organization.

## 2.2. Sample size justification

Note that the sample size of 60 participants was somewhat arbitrarily yet conservatively set in order to compensate for potential drop-outs. For example, a power analysis to detect a medium-sized effect ($d = 0.50$) between 'equal description versus experience' trials within one context condition (e.g. 20% filled bar and emoji with a 20% gain probability) would have suggested that we have over 80% (Experiment 1) and over 99% (Experiment 2) power to detect a significant effect (one-sided) with a significance level of 5%.

## 2.3. Apparatus and stimuli

Except as otherwise described, the apparatus and stimuli were identical in the two experiments. The experiment took place online and stimulus presentation and recording of responses were controlled by jsPsych [6]. A centrally positioned '+' on a light grey background served as a fixation point. As can be seen from figure 1b, the *description stimuli* were partially filled with grey rectangles. The idea behind these stimuli is that the 'filledness' of the stimuli concretely describes the associated payoff/loss

probability (for a similar approach, see [4,5]). Participants were not explicitly told about the filledness concept and needed to learn it in the experimental session. Specifically, the rectangles were either 20% filled (thus 80% grey) to describe a payoff/loss probability of 20%. Similarly, 50%-filled and 80%-filled rectangles described payoff/loss probabilities of 50% and 80%, respectively. The fill colour was always blue in Experiment 1. The fill colour in Experiment 2 was green during the gain context and red during the loss context.

We used emojis as the *experience stimuli*. The idea behind these stimuli was that, in contrast to the descriptive bar stimuli, no feature of these images could be readily be interpreted as a continuous scale of probability. To counteract any possible confound of image familiarity or preference, we randomly selected three emojis from a library of 34 for each participant and randomly allocated gain/loss probabilities of 20%, 50% and 80% to the selected emojis. This randomization ensured that experience-related biases in choice patterns could not be explained as a systematic preference for any specific image. Novel emojis were used across contexts in Experiment 2—that is, a new set of three randomly selected emojis were used in the second context participants experienced in their session.

## 2.4. Procedure

The procedure was similar between the two experiments except that context was manipulated between-subject in Experiment 1 but within-subject in Experiment 2 and top-scoring participants could win a 10€ voucher in Experiment 2 but not in Experiment 1. Furthermore, participants in Experiment 1 were asked about their subjective inferences of the probabilities associated with the different stimuli after the experiments (procedural details are presented in the corresponding results subsection).

### 2.4.1. Experiment 1

Each participant completed eight blocks of 96 trials (figure 1a). Prior to the start of the first (training) block, participants were instructed to select the best option to either earn points or to mitigate losses (depending on whether they participated in the gains/loss-oriented context). In order to more fairly compare how participants learned about the description and experience options, participants were not told that the description stimuli were explicitly probability cues. Participants were only instructed to finish their session with the greatest number of points possible. Participants in the gains-oriented context began each session with 0 points and could earn points, whereas participants in the loss-oriented context began with 600 points and could only lose points. All other aspects of the procedure were identical between the two experiments. Breaks between blocks were self-paced and participants could always see their current score. The first two blocks were training blocks where participants made decisions either exclusively between description (partially filled bars) or experience (emojis) stimuli. The purpose of these blocks was to introduce participants to the task and stimuli types and to ensure that they understood the task. Half of the participants were first tested in the 'pure-description' training block, followed by the 'pure-experience' training block (order counterbalanced for the other half of the participants). After the training blocks, participants completed six experimental blocks in which they were randomly presented with decisions between (i) two descriptive (bar) stimuli of different gain/loss probabilities, (ii) two experience (emoji) stimuli with different gain/loss probabilities, (iii) both a descriptive and experience option with different gain/loss probabilities and (iv) both a descriptive and experience option with equal gain/loss probabilities. See figure 1b for examples of each of these trial types.

### 2.4.2. Experiment 2

Each participant completed 10 blocks of 96 trials (figure 1a). Each participant experienced both the gain and loss context in a single approximately 40 min session. Half of the participants first experienced the gain context followed by the loss context (order counterbalanced for the remaining half of participants). Both contexts began with an information screen telling participants to either collect as many points as possible or to mitigate the loss of points. Importantly, participants were informed that the top 10 scoring participants would win a 10€ shopping voucher. Following the information screen, participants completed two training blocks in a manner identical to Experiment 1 followed by three experimental blocks that were identical in composition to Experiment 1. Following completion of the fifth block, participants were informed that the context changed and that they should now either seek gains or mitigate losses (depending on what the first context was in their session). This new

information screen was followed by two new training blocks which were then followed by three experimental blocks of 96 trials identical in composition to the experimental blocks in the first half. New stimuli were used in the second context a participant experienced. Specifically, three new emojis were selected as experience stimuli and bar stimuli with a different fill colour were used (red fill for the loss context and green fill for the gain context).

## 2.5. Description and experience training blocks

Participants began trials by fixating on a central cross for 500 ms and then were presented with two different stimuli (emojis in the 'experience' block; partially filled bars in the 'description' block) of the left/right side of the computer screen. The stimuli remained on the screen until participants responded, and no constraint was placed on participant's choice reaction times (RTs). Responses were made with the left and right index fingers by pressing the 'Q' (left) and 'P'-keys (right). Following a choice, the screen was extinguished and the fixation cross was re-presented for 500 ms. Following this delay, feedback was presented for 500 ms. In the gains-oriented context, a '+1' indicated that the trial was rewarded and a '+0' indicated it was not. In the loss-oriented context, a '−1' indicated a trial lost points and a '−0' indicated no loss.

## 2.6. Main, experimental blocks

Individual trial sequences during the main blocks were identical to the training blocks. Four equally occurring trial types comprised the main blocks—'pure-description' trials where participants decided only between description stimuli (identical to the description training block); 'pure experience' trials where participants decided only between experience stimuli (identical to the experience training block); unequal description versus experience trials where participants were presented with both types of stimuli but one was 'better'; equal description versus experience trials where participants were presented with equiprobable description and experience options. All stimulus combinations were equally presented and counterbalanced by side (that is, all stimuli were presented an equal number of times on both the left and right sides of the computer screen).

## 2.7. Gain/loss-oriented contexts

In the gain-oriented context, participants began each session with zero points and were instructed to gain points by selecting the probabilistically 'better' of two simultaneously presented options. Choice-option probabilities corresponded to the likelihood of a selected stimulus yielding a point; thus, the 'better' options in the gain context were the higher probability ones.

In the loss-oriented context, participants were instructed to retain as many points as possible by selecting the probabilistically 'better' of two simultaneously presented options. Choice probabilities in this context corresponded to the likelihood of a selected stimulus subtracting a point; thus, the 'better' options in this context were the low-probability ones. Participants in Experiment 1 began a loss-context session with 600 points; participants in Experiment 2 were given 600 points when they entered the loss context so that they could retain the points gained during the gain context (that is, they could only lose the new points in the loss context).

## 2.8. Data preparation

For all analyses, we excluded trials with outlier choice RTs. Following visual inspection of the data, we defined outlier RTs as those less than 100 ms and greater than 3000 ms. These excluded trials comprised 3.51% of the data in Experiment 1 and 3.65% of the data in Experiment 2. We explored alternative outlier criteria (e.g. less than 50 ms and greater than 4000 ms) and obtained qualitatively similar results. We performed no other data preparation prior to formal analysis.

## 2.9. Data analysis

Our primary-dependent measures were choice patterns (e.g. $p$(Choose better) and $p$(Choose Experience)) and choice RTs. For Experiment 1, we primarily used mixed repeated-measures ANOVAs where context was a between-subject factor and choice condition and, when applicable, stimulus type, were within-subject factors. For Experiment 2, we primarily used repeated-measures ANOVAs where all

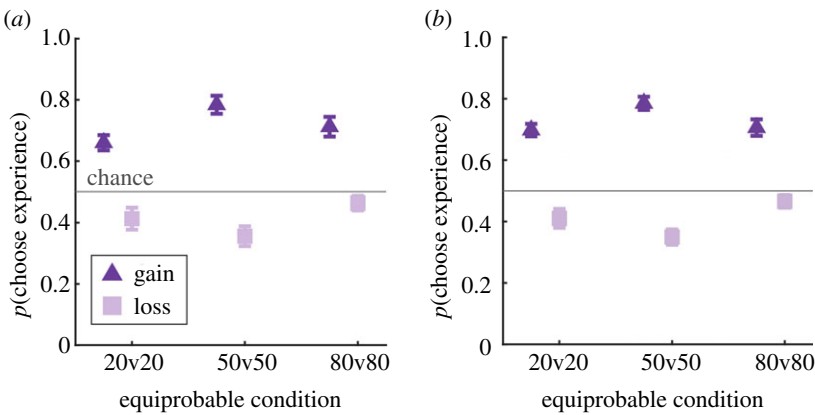

**Figure 2.** Choice patterns during the equiprobable description versus experience trials. Data are grouped according to context and probability. (a) Data from Experiment 1 where context was a between-subject factor. (b) Data from Experiment 2 where context was a within-subject factor. Markers indicate group means ± s.e.m.

factors were within-subject. Unless explicitly stated otherwise, the analysis of Experiment 2 was pre-registered on the Open Science Framework. As can be seen in our preregistration, our core questions for Experiment 2 were similar to Experiment 1. All MATLAB code used to analyse the data are available on our Open Science Framework page: https://osf.io/a8zf6/?view_only= b43e278fe15d4fe5a51dd52ee4bc9c30.

# 3. Results

## 3.1. Core results—choice patterns during 'equal descriptive versus experience' trials

To test the central predictions of a classic DE gap (figure 1c), we compared the proportion of times participants selected the experience options during the equiprobable trials as a function of probability and context separately for each experiment. In both experiments, gain-seeking participants in both experiments generally preferred the experience stimuli across all probabilities whereas loss-mitigating participants generally preferred the description stimuli across all probabilities (figure 2). For Experiment 1, where context was a between-subject factor, we quantified this effect via a mixed repeated-measure ANOVA with the within-subject factor of equiprobable condition (20v20 versus 50v50 versus 80v80) and a between-subject factor of the experimental context. This analysis revealed significant main effects of probability condition ($F_{2,116} = 3.59$, $p = 0.03$, $\eta_p^2 = 0.06$, context ($F_{1,58} = 72.55$, $p < 0.001$, $\eta_p^2 = 0.56$), and their interaction ($F_{2,116} = 14.45$, $p < 0.001$, $\eta_p^2 = 0.20$). We obtained a qualitatively similar result pattern for Experiment 2, where context was a within-subject factor and data were assessed with a repeated-measures ANOVA (probability: $F_{2,59} = 1.63$, $p = 0.20$, $\eta_p^2 = 0.05$; context: $F_{1,59} = 109.05$, $p < 0.001$, $\eta_p^2 = 0.65$; probability*context: $F_{2,59} = 16.97$, $p < 0.001$, $\eta_p^2 = 0.37$). These results indicate that participants preferred the experience options across all probabilities when seeking gains and, conversely, preferred the description options across all probabilities when mitigating losses.

Next, we asked which probability would show the greatest DE gap in each of the two experiments. We evaluated this by running repeated-measures ANOVAs for each context in each experiment separately and then comparing DE gap size as a function of probability via *post hoc* multiple comparisons. These analyses revealed the same pattern of ANOVA factor results as the repeated-measures analyses and the multiple comparisons tests indicated that the maximal DE gap occurred at 50%. Specifically, gain-seeking participants in both experiments showed the largest DE gap at 50% (all $p < 0.05$). Similarly, loss-mitigating participants in both experiments demonstrated the largest DE gap at 50% (Exp 1: EQ20 versus EQ50 $p = 0.07$, EQ50 versus EQ80 $p < 0.001$; Exp 2: EQ20 versus EQ80 $p = 0.07$, EQ50 versus EQ80 $p < 0.001$). Thus, contrary to the result pattern predicted by prior literature [1,4,7], our equiprobable choice results suggest that the DE gap arises as a function of context and is maximal at 50%.

## 3.2. Control analyses—task comprehension and choice pattern stability

To adequately assess the DE gap in our paradigm, it was critical that participants form relatively accurate and stable inferences of the payoff/loss probabilities associated with each stimulus. This would ensure

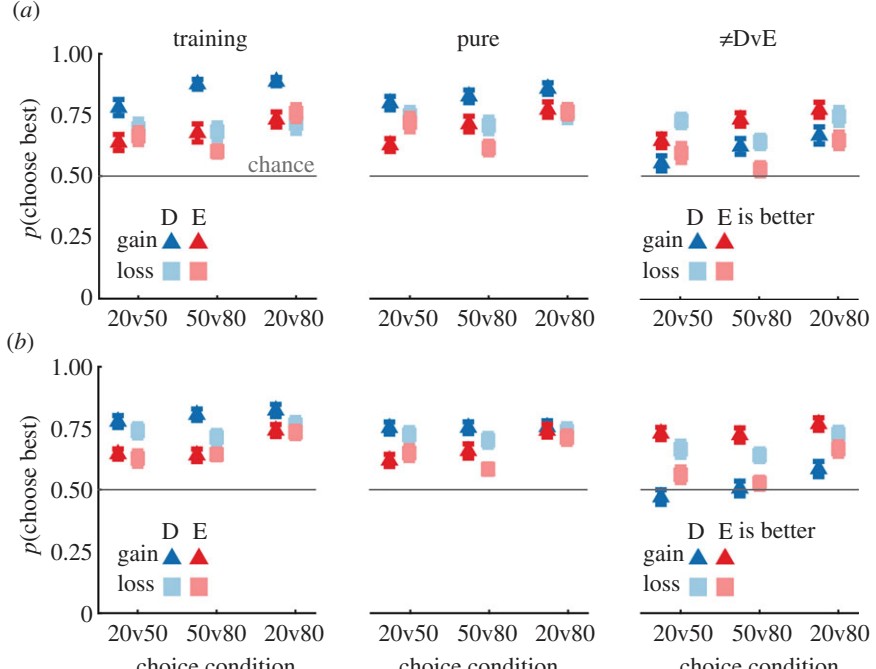

**Figure 3.** Choice patterns of Experiment 1 (*a*) and Experiment 2 (*b*) grouped by context and stimulus type during the training blocks, main block pure-description/experience trials and unequal description versus experience trials. Markers indicate group means ± s.e.m. See table 1 for summaries of the corresponding ANOVAs.

that our core analyses concerning the equiprobable gambles reflect decisions driven by formed, relatively stable probability representations and cannot be explained as artefacts of a learning process, sampling error or large fluctuations of inferred value across the session. Thus, our general hypothesis here was that participants would demonstrate preferences for the probabilistically 'better' option during trial conditions containing an objectively 'better' option (i.e. during the training blocks, the 'pure descriptive/experience' and the 'unequal descriptive versus experience' trials during the main blocks).

### 3.2.1. Training blocks

The 'Training' column in figure 3 ((*a*): Exp 1, (*b*): Exp 2) shows the mean proportion of participants choosing the 'better' option in the training blocks as a function of choice condition (20v50, 20v80, 50v80), training/stimulus type (experience, descriptive) and experimental context (gain, loss). The mean proportion was larger than 50% for all conditions across both experiments (all $p < 0.001$, $t$-tests against 0.5; all Cohen's Ds > 0.9), indicating that participants acquired the task.

For completeness, we also assessed the probability of choosing the 'better' option as a function of our experimental factors (Choice Condition, Stimulus Type and Context). We used mixed repeated-measures ANOVAs to assess data from Experiment 1 and repeated-measures ANOVA to assess data from Experiment 2. These results (see table 1 for precise details and effect sizes) indicate that participants in both experiments were more likely to select the better option when making choices between the description stimuli as compared to the experience stimuli, suggesting that the description stimuli were easier for participants to learn. We also found that choice accuracy increased as the difference between the options increased (e.g. peak accuracy at 20v80). These effects were most pronounced in the gain context, suggesting that participants may have found the loss context more difficult.

### 3.2.2. Pure trials during experimental block

The 'pure' column in figure 2 shows the choice patterns in 'pure' trials during the main blocks were similar to that of the training blocks. We confirmed the stability of this choice pattern via an identical analysis to that of the training blocks. Separate one-sided $t$-test against chance level (50%) revealed again significant effects for all conditions (all $p < 0.001$, all Cohen's D > 0.75), indicating that participants' task comprehension was retained throughout the main experimental blocks.

**Table 1.** Summary of control choice pattern analyses of variance (figure 3). NB. For ≠DvE Choice, Type refers to Better Stimulus Type. Mixed repeated-measures ANOVAs were used for Exp 1; repeated-measures ANOVAs were used for Exp 2.

| Exp | effect | d.f.1, d.f.2 | training choice patterns (figure 3) | | | experimental choice patterns (figure 3) | | | ≠DvE choice patterns (figure 3) | | |
| --- | --- | --- | --- | --- | --- | --- | --- | --- | --- | --- | --- |
| | | | Fstat | Pval | $\eta_p^2$ | Fstat | Pval | $\eta_p^2$ | Fstat | Pval | $\eta_p^2$ |
| 1 | context | 158 | 7.39 | <0.01 | 0.11 | 1.76 | 0.19 | 0.03 | 0.45 | 0.50 | <0.01 |
| 1 | condition | 2116 | 12.23 | <0.001 | 0.17 | 20.37 | <0.001 | 0.26 | 19.63 | <0.001 | 0.25 |
| 1 | stimulus type | 158 | 14.89 | <0.001 | 0.20 | 34.12 | <0.001 | 0.37 | 0.04 | 0.84 | <0.01 |
| 1 | context * condition | 2116 | 4.81 | <0.01 | 0.08 | 13.84 | <0.001 | 0.19 | 14.75 | <0.001 | 0.20 |
| 1 | context * type | 158 | 8.45 | <0.01 | 0.12 | 9.62 | 0.003 | 0.14 | 15.84 | <0.001 | 0.21 |
| 1 | condition * type | 2116 | 4.66 | 0.01 | 0.074 | 4.27 | 0.02 | 0.07 | 0.43 | 0.65 | <0.01 |
| 1 | context * cond * type | 2116 | 1.45 | 0.19 | 0.03 | 3.11 | 0.05 | 0.05 | 1.0 | 0.50 | 0.02 |
| 2 | context | 159 | 4.11 | 0.05 | 0.07 | 3.11 | 0.05 | 0.05 | <0.01 | 0.97 | <0.01 |
| 2 | condition | 259 | 14.67 | <0.001 | 0.33 | 1.76 | 0.19 | 0.03 | 22.29 | <0.001 | 0.43 |
| 2 | stimulus type | 159 | 24.37 | <0.001 | 0.29 | 20.37 | <0.001 | 0.26 | 0.04 | 0.84 | <0.01 |
| 2 | context * condition | 259 | 0.26 | 0.77 | <0.01 | 3.89 | 0.02 | 0.12 | 1.92 | 0.15 | 0.06 |
| 2 | context * type | 159 | 1.96 | 0.17 | 0.03 | 0.04 | 0.84 | <0.01 | 0.38 | 0.68 | <0.01 |
| 2 | condition * type | 259 | 4.28 | 0.02 | 0.13 | 8.32 | <0.001 | 0.22 | 32.94 | <0.001 | 0.36 |
| 2 | context * cond * type | 259 | 1.45 | 0.24 | 0.05 | 1.78 | 0.17 | 0.06 | 2.38 | 0.10 | 0.08 |

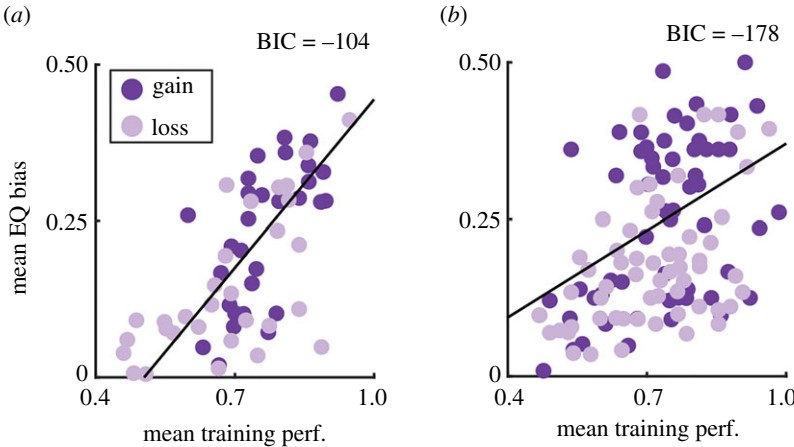

**Figure 4.** Overall training performance predicts the size of individual participants' DE gaps during equiprobable trials. The x-axes depict the participants' mean performance across all training trials. The y-axis depicts participants' mean equiprobable bias (absolute difference from 50%) across all probability levels during the equiprobable trials during the main blocks. The model integrating both types of training blocks was the best fit (with the lowest BIC scores) across both the Experiment 1 (*a*) and Experiment 2 (*b*). BIC = Bayesian information criterion, a measure of model fit used for model comparison. See electronic supplementary material, figure S2 for alternative models fit to only the Description and Experience training blocks. See electronic supplementary material, figure S3 for data grouped by individual probabilities.

As with the training choice patterns, we again assessed choice performance as a function of Choice Condition, Stimulus Type and Context. These results, described in detail in table 1, were qualitatively similar to the training results, indicating that choice patterns were stable throughout the experimental blocks. Furthermore, as a non-pre-registered analysis, we explored whether the outcomes of selecting a given option on one trial influenced decisions on subsequent trials that option was present. Specifically, we conducted a win-stay, lose-shift analysis which indicated that participant choice patterns were stable throughout the sessions (see electronic supplementary material, figure S1 and Data for details).

### 3.2.3. Generalization of inferred probabilities across stimulus types

To see whether participants could generalize and accurately compare inferred probabilities from the two stimulus types, we assessed the distribution of participant $p$(Choose Better Option) during the 'unequal descriptive versus experience' trials. We predicted that participants would significantly prefer the 'better' option regardless of which stimulus type is 'better' on a given trial (see ≠DvE column in figure 3).

We assessed this via a mixed repeated-measures ANOVA for Experiment 1 and a repeated-measures ANOVA for Experiment 2 considering whether and how $p$(Choose best) varied with the factors probability condition (20v50, 20v80, 50v80), context (gain, loss) and 'best' stimulus type (i.e. whether emoji or bar was the 'better' stimulus on that trial). These analyses, detailed in table 1, revealed that participants were more likely to choose the better option when the difference between the options was greater (e.g. 20v80) and that this effect was reduced in the loss context, consistent with the notion that participants in the loss context found their task more difficult.

Interestingly, participants in the gain context were more likely to select the 'better' option when it was an experience option and that this bias was reversed towards the descriptive bar stimuli for participants in the loss context. These results fit well to the findings of the equiprobable trials (figure 2) in which the preference for experience stimuli was also modulated by experimental context (gain-seeking participants preferred the experience options during equiprobable gambles, whereas loss-mitigating participants preferred the description options).

## 3.3. Complementary and exploratory analyses

Although our experimental goals ultimately depended on participant choice patterns during the 'equal descriptive versus experience' trials, we were interested in whether we could explain those effects in terms of initial learning (figure 4; see also electronic supplementary material, figures S2 and S3). In further exploratory analyses, we examined whether any of the context and stimulus type-related

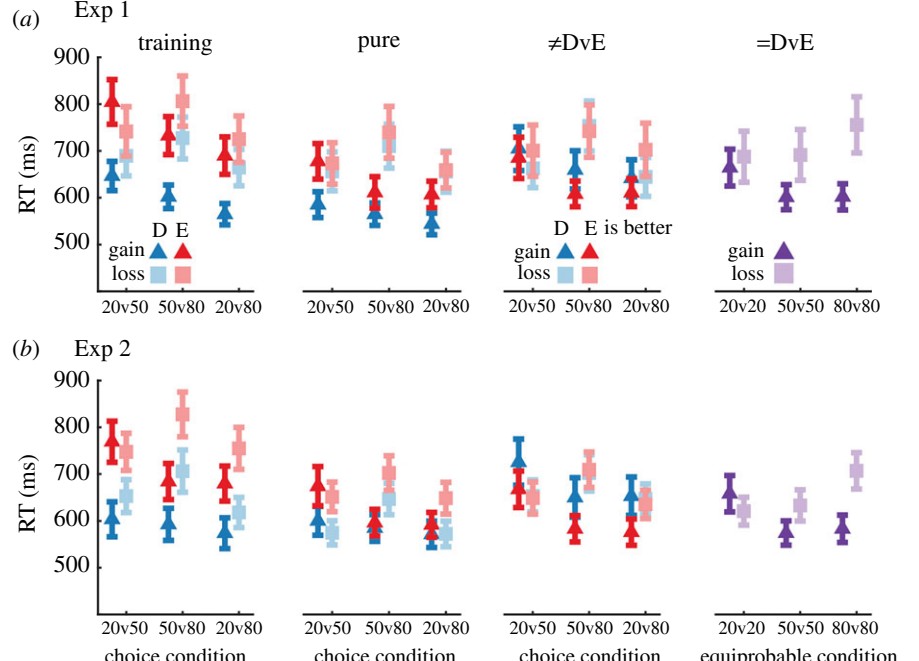

**Figure 5.** Participant choice RTs grouped by context and stimulus type during training blocks, main block pure-description/experience trials, ≠DvE trials, = DvE trials. Markers indicate participants means ± s.e.m. See electronic supplementary material, tables S1 and S2 in the electronic supplementary material for summaries of the corresponding ANOVAs.

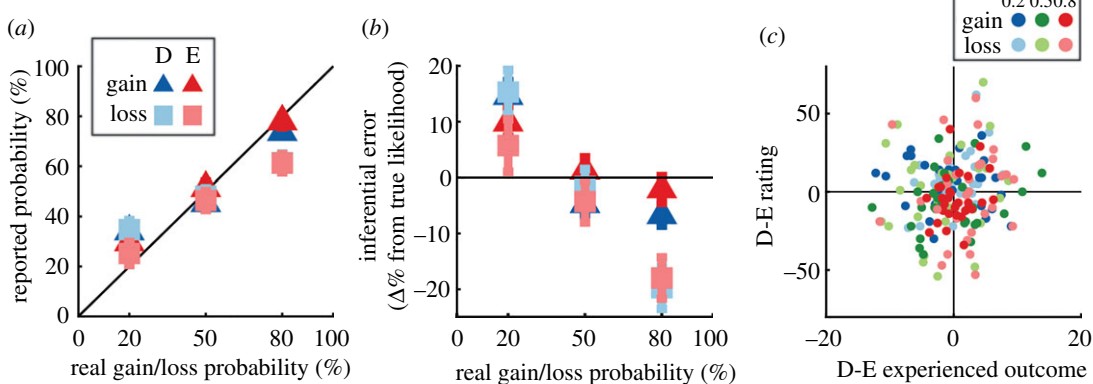

**Figure 6.** Subjective inference surveys. Following the completion of the between-subject experiment, participants were asked to explicitly report the outcome probabilities for each of the six stimuli used in their session; (*a*) shows the raw reported probabilities as a function of how often each option yielded a non-zero outcome; (*b*) shows how different those raw reports were from the true outcome probability of each stimulus; (*c*) shows that differences in subjects' reported probabilities for equiprobable description and experience options do not vary as a function of the difference in how often those same stimuli yielded non-zero outcomes. In other words, participants' ratings of the likelihoods in the survey did not correspond to either their actual choice patterns or the hundreds of outcomes of those choices. Markers in (*a*) and (*b*) depict group means ± s.e.m. Markers in (*c*) denote individual participant means for each stimulus type and probability level.

choice biases we observed would show corresponding changes in choice RT (figure 5) and metacognitive awareness (figure 6, only data from Experiment 1). In an additional, non-registered analysis conducted after the collection of all data, we explored whether these choice patterns could be recovered in a reinforcement learning framework (electronic supplementary material, figure S4).

### 3.3.1. Initial learning predicts the magnitude of stimulus type-related probability distortion

Having found clear effects of stimulus type on participant choice patterns during the equiprobable 'description versus experience' trials (figure 2), we were interested in whether the magnitude of

individual participants' equiprobable choice biases could be explained in terms of how they initially inferred values ($p$(Outcomes)) of the stimuli.

We explored this by fitting and comparing several linear mixed-effects models where participants' mean choice bias during the equiprobable 'description versus experience' trials was predicted as a function of training performance and context (figure 4; electronic supplementary material, figures S2 and S3). Equiprobable choice bias was calculated as each participant's unsigned mean difference from 50% across all equiprobable gambles. The best-fitting model across both experiments, as determined by a significant $p$-value and the lowest Bayesian information criterion, was the one which integrated training performance across both the description and experience training blocks (figure 4). Specifically, we used the mean of both training blocks, thereby integrating across both, as a predictor. Significant, positive coefficients we obtained for the training performance factor across both experiments (Exp 1: $\beta = 0.89$, $p < 0.001$; Exp 2: $\beta = 0.46$, $p < 0.001$). No other factors were significant, indicating that the initial inferences formed for both stimuli types influence probability perception and that this inference-driven distortion occurred similarly in both the gain and loss contexts (see electronic supplementary material, figures S2 and Data for details of other candidate models).

Because we found that individual participants' DE gaps were related to their initial learning, we wondered whether our primary results in figure 2 entirely depended on participants' initial learning. We addressed this with two analyses: first, we re-ran the analysis underlying figure 2 and included training performance as a covariate. This analysis revealed the same pattern of effects as the original analyses, indicating that while training performance may play a role in establishing the probability distortions underlying the DE gap, these distortions are larger and more stable than the effects of initial learning. The second analysis involved re-running the regressions underlying figure 4 for each probability separately (see electronic supplementary material, figure S3). This analysis allowed us to ask whether the DE gap observed at each probability level depended on training performance for each of those probabilities (in case one probability was more difficult to learn than another). This analysis did not reveal a main effect of probability (see electronic supplementary material, Data for details), indicating that the DE gap results we observed in figure 4 cannot be explained by differences in learning across the different probabilities.

### 3.3.2. Reaction times

Given the strong probability and context related effects we observed in participant choice patterns, we wondered whether the dynamics of those decisions, as measured by RTs, would show similar modulations. Similar to the choice data, we assessed the data from Experiment 1 with mixed repeated-measures ANOVAs and assessed the data from Experiment 2 with repeated-measures ANOVAs. These data are plotted in figure 5. For concision, we report the exact details of these ANOVAs in electronic supplementary material, tables S1 and S2.

RT patterns were qualitatively similar across both experiments. During the training and pure-description/experience blocks, participants responded faster when choosing between description options as compared to experience ones and were generally faster when the difference between the options was greater (e.g. in the 20v80 condition). Furthermore, participants were required longer to decide in the loss context as compared to the gain context. These results mirror the choice patterns, suggesting that participants better understood the description stimuli and may have found the loss condition comparatively more difficult.

Next, we explored RTs during the unequal 'descriptive versus experience' trials. Mirroring the training and pure experimental block RT patterns, participants were generally faster to respond when the difference between the options was larger (e.g. 20v80) but this effect was reduced context when participants mitigated losses as compared to when participants sought gains. Furthermore, gain-seeking participants were generally faster when the experience stimuli were the 'better' options and this pattern was reversed when participants mitigated losses—their choices were faster when the description stimuli were the 'better' options. These results broadly parallel the choice pattern results in these conditions.

Finally, we assessed how RTs during the 'equal descriptive versus experience' trials varied by context and equiprobable condition. We found that participants in the loss context took longer to respond as the probability of losing a point increased but that participants in the gain context responded similarly regardless of the payoff likelihood. Considering the difference between these conditions, one interpretation is that participants in the loss context experienced greater choice conflict as they were presented with increasingly unfavourable options.

### 3.3.3. Metacognitive awareness

After completing the main experimental blocks, participants in Experiment 1 completed a brief survey where they reported their subjective beliefs about the payoff/loss probabilities associated with the stimuli used in their session. For example, if a given subject's stimuli set consisted of a 20% filled bar, a 50% filled bar, an 80% filled bar, a dragon emoji, a sun emoji and a fish emoji, the participant would be, for example, shown the dragon emoji and then asked to adjust a slider on the screen to the per cent payoff/loss likelihood they believed it represented. This component was omitted in Experiment 2 so that explicit judgements regarding one context would not influence the subsequent context.

Participants' mean subjective ratings for each stimulus are presented in figure 6a. We assessed the inferential error plotted in figure 6b using a mixed repeated-measures ANOVA. Inferential error was defined as the difference in participants' ratings and the proportion of times selecting a given option yielded a gain or loss, respectively. This analysis detected significant effects of context ($F_{1,58} = 7.55$, $p = 0.008$, $\eta_p^2 = 0.11$), probability ($F_{2,116} = 44.30$, $p < 0.001$, $\eta_k^2 = 0.43$) and their interaction ($F_{2,116} = 4.65$, $p = 0.01$ $\eta_p^2 = 0.074$). This indicates that participants reported low-probability described options as more likely to occur than equiprobable experience ones, the reverse at high probabilities, and that the effect was more pronounced for loss-mitigating participants than gain-seeking ones. These results are consistent with the classic description-oriented DE gap but, interestingly, when comparing this result pattern against their choice patterns in the = DvE condition (figure 2), participants' ratings appeared orthogonal to their actual choice behaviour.

Next, we assessed whether the difference in participants' ratings of description and experience stimuli were related to how often they experienced non-zero outcomes (that is, gains or losses, respectively). Similar to the inferential error scores underlying figure 6b, we asked whether the paired differences between equiprobable description and experience stimuli ratings were correlated with the paired differences in the proportion of times those equiprobable stimuli yielded a gain or loss. We used a linear mixed-effects model to probe whether such a relationship would vary as a function of context and probability. The results of this analysis, presented in figure 6c, revealed no systematic relationship between ratings differences and outcome likelihood differences for paired equiprobable description and experience stimuli. No factors in the model returned significant (all $p > 0.65$). This result indicates that participants' metacognitive beliefs regarding the probabilities were independent of their real-choice behaviour and the exact patterns of outcomes they experienced.

## 4. General discussion

At its core, the DE gap indicates that probabilistic perception is influenced by whether values of choice options are presented descriptively/symbolically or whether those values were inferred through repeated experience [1,7]. The classic DE gap profile is that of a stimulus type*probability interaction (e.g. low-probability described options are perceptually inflated relative to experience ones and the reverse at high probabilities). Unlike prior paradigms which deduced participant probability perceptions by comparing their choice patterns against a reference 'sure-bet' option, we measured the DE gap directly by requiring participants to decide between described and experience stimuli with equal outcome probabilities in different contexts (gain and loss). This allowed us to directly assess whether the probability distortions classically ascribed to the DE gap can be directly observed without any intermediate steps and whether/how such distortions are modulated by the prospect of gain or loss.

Contrary to expectations, we found no evidence that the DE gap's size and orientation was linearly related to probability. However, we did find that gain-seeking participants significantly preferred the experience stimuli over equiprobable described options at all probability levels and that this pattern was reversed for loss-mitigating participants (who preferred described options at all probabilities). Furthermore, we found a maximal effect at 50% across both contexts in both a between-subject and within-subject manipulation of context. These results complement reports the DE gap is experience-oriented and is not limited to rare (low probability) events [5]. In the following, we rule out alternative explanations against the background of prior literature and discuss the implications of these findings for models of probability perception as a function of probability and context.

### 4.1. Relation to prior literature

We were surprised to not observe choice patterns consistent with classic reports of the DE gap [1]. Prior reports (reviewed in [3]) of the DE gap found that its size and profile is influenced by learning and

sampling error, suggesting that insufficient learning influences the size of the gap (i.e. greater sampling error leading to a larger gap). However, this is unlikely in our dataset because participant choice patterns during the training and 'pure-description/experience' main experimental blocks indicated that participants accurately inferred the hierarchy of probabilities. Furthermore, exploratory analyses indicated that individual participants' training performance across both training blocks was positively related to the size of individual participants' DE gaps. This result indicates that participants' initial inferences of option outcome probabilities influenced their individual DE gaps in a manner opposite to what might be expected from sampling error driving the gap. Because other studies of human decisions under risk have found substantial differences in choice patterns depending on whether participants were financially incentivized or not [8–10], it is noteworthy that we replicated this pattern both with and without financial incentive which suggests that there is a negligible effect of financial incentive in our paradigm.

It should be emphasized that prior studies distinguished described and experience choice options via language (e.g. [1,7]) and it was unclear whether the DE gap could be observed in a situation where both described and experience options were non-verbal. To compare experience and description options more fairly, we used non-verbal descriptive stimuli without telling participants about the 'filledness'-concept. Our idea here was that descriptive options are fundamentally more informative ones which would be learned as a symbolic system and therefore more exploitable as compared to the entirely uninformative experience options (emojis). Thus, it is important to emphasize that the fundamental idea underlying the description and experience stimuli in our paradigm is *informativeness*. Indeed, our participants' more optimal choice behaviour and faster RTs when deciding between description (as compared to experience) options suggests that they did learn this symbolic system and that it aided their decision-making. Furthermore, our results cannot be explained by a simple preference for emojis because participants differentially preferred them when seeking gains and avoided them when mitigating losses.

Most important, we investigated the characteristics of the DE gap without using a 'sure-bet' option (as was typically used in prior studies) but instead by directly comparing choice patterns between description and experience options with equal outcome probabilities. This is an important feature of our experimental design because it means that we did not compute any CEs, and we did not collapse across multiple probabilities when assessing our data. That we did not observe choice patterns predicted by early reports of the DE gap addresses one of our core questions of whether the stimulus type*probability DE gap profile depends on including 'sure-bet' options, computing CEs and, more generally, collapsing across many trial conditions. Like us, Ludvig & Spetch [5] found an effect at 50% by looking at raw choice patterns. It should not be understated that this earlier finding was surprising because early formulations of the DE gap defined it precisely according to this stimulus type*probability interaction such that the gap was maximal at low (less than 30%) probabilities and nearly zero at 50%. Considering that we found the DE gap to be maximal at 50%, this again raises the possibility that the mechanisms underlying the DE gap are highly sensitive to differences in how choices are contextualized.

Our findings integrate and extend Heilbronner & Hayden's [4] study of the DE gap in which gain-seeking NHPs were faced with equiprobable decisions. Like our gain-seeking human participants, Heilbronner & Hayden's [4] NHPs appeared to prefer the experience options at all probability levels (albeit most strongly at low probabilities). However, Heilbronner & Hayden [4] never directly compared choice patterns exclusively within equiprobable trials, instead of collapsing across all trial conditions which contained an experience option of a certain probability, making it difficult to draw firm conclusions from a comparison.

Notably, however, many prior studies of the DE gap had found a description-orientation, meaning that the likelihoods associated with low-probability description options (as compared to experience ones) were perceptually inflated (e.g. [3,7]). For example, Abdellaoui *et al.* [7] studied the DE gap across a range of probabilities in the contexts of gains and losses and found a description-oriented effect that was most pronounced at low probabilities when seeking gains and a small-to-negligible effect when participants were faced with the prospect of loss. Interestingly, these choice patterns were fundamentally different from what we observed (strong experience-oriented DE gaps across all probabilities with the large effect of context). In addition to the potential influence of language (see above), we speculate that serialization and the intervals between choices may be key factors contributing to this discrepancy.

Like early studies of the DE gap, Abdellaoui *et al.* used relatively few choices (less than 14) with uncontrolled intervals between those choices. By contrast, our design involved participants making

hundreds of choices with very short intervals between them. Serialization has long been appreciated to promote risk-seeking (for gains) because it allows participants to amortize the risk associated with each choice [11,12]. Short intervals between choices have also been implicated in modulating risk-seeking in NHPs [13], which can be interpreted as a form of temporal discounting which is known to be abnormal in gambling-addicted humans [14,15]. Considering that studies like ours and others [4,5] which used serialized choice report experience-oriented effects and studies which did not use serialized choice (e.g. [1,7]) report description-oriented effects, future studies are needed to precisely discern whether and how serialization mediates the DE gap's orientation.

In sum, our results indicate that the stimulus type*probability characterization of the DE gap in its classic form may be less omnipresent than anticipated. Instead, our findings complement reports [4,5] that the DE gap's direction, at least within the context of serialized choice, may be experience oriented and that the DE gap is not limited to rare (low probability) events. In the following, we propose how the present result can be conceptualized within a simplified modelling architecture of probabilistic inference.

## 4.2. Towards a probabilistic inference model of probability perception

Broadly, risk attitudes describe how decision-makers perceive risk given different contexts [16,17]. In line with serialization prompting risk-seeking for gains [11,12,18], our participants' preference for experience stimuli when seeking gains suggests that experience options were perceived as riskier than description options. Similarly, were the experience options perceived as riskier, that participants avoided them when mitigating losses could be explained as a form of uncertainty aversion [19–21]. Thus, our context manipulation seems to have elicited different risk attitudes that varied in terms of the uncertainty associated with the description/experience options. In the following paragraphs, we offer a theory of how probability distortions might arise through the incorporation of uncertainty during the initial inference/learning of the outcome likelihoods associated with description and experience options.

Although often used interchangeably, risk and uncertainty refer to two different but related concepts. Risk characterizes settings with known probabilities and uncertainty describes settings without known probabilities [22]. More fundamentally, from a behavioural modelling perspective, uncertainty connotes the processes of learning and inference, whereby uncertainty about a choice option is reduced through repeated experience [21,23]. One possibility is that the experience-oriented DE gap we and others observed could be explained in terms of different degrees of uncertainty associated with described and experienced options [24].

To illustrate the idea, consider that associations between a choice and its consequence are learned by comparing an expected outcome against an experienced one [25,26]. Thus, differences in the uncertainty associated with a choice option could lead to different inferences about its value. For example, participants inferring the payoff probability of an informative choice option (i.e. a stimulus whose features describe the payoff probability) might experience less uncertainty with each outcome (because the stimulus explicitly describes to the subject what to expect). When using outcomes of such descriptive/informative options to infer the outcome likelihoods, participants might ultimately infer sharper, more accurate inferences (similar to inferring a probability density function (PDF)). By contrast, when participants infer the outcome likelihood of an 'uninformative' choice option, such as the experience stimuli in the present experiment, there may be more uncertainty associated with each outcome (because nothing about the stimulus cues the subject what to expect). This could have led participants to form broader, noisier inferences of the outcome likelihoods of the uninformative, experience options as compared to equiprobable described options. We schematize the result of such a process in figure 7 (the size of the circles is highlights different amounts of uncertainty associated with the outcomes of choices from description and experience with each circle denoting one outcome; larger circles indicate more uncertain outcomes). In the next paragraph, we theorize how such differences in outcome-mediated inference could lead to distortions of probabilistic perception.

Fundamentally, we theorize that participants may infer the likelihood of outcomes in a manner similar to a PDF—that is, participants may form internal beliefs about the probability that a given probability is the true outcome likelihood *vis a vis* estimating the weightedness of a die or coin. We could express one point in that distribution as the posterior probability $p(\text{Probability} = Xp \mid \text{Data})$ where $Xp$ could be any number between 0 and 1, and the distribution would be the set of all points between 0 and 1. Such a process is compatible with and extends the findings from our distributional reinforcement model (see electronic supplementary material, figure S4 and details), which suggests that participants learned different values for the experience and description options due to differences

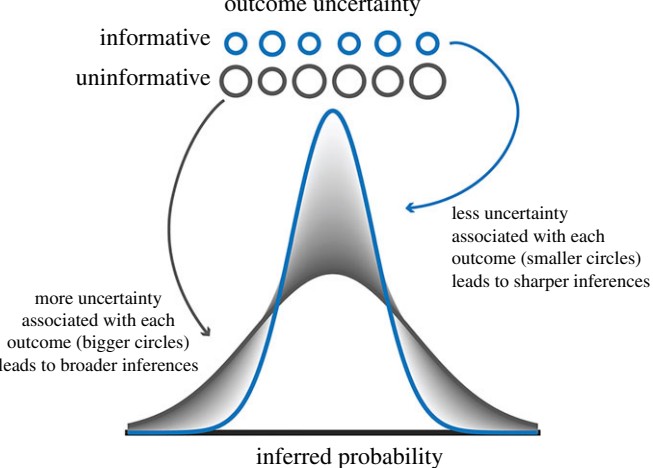

**Figure 7.** Graphical depiction of uncertainty-driven differences in inferential processing associated with informative and uninformative choices. The top row schematizes the uncertainty associated with each outcome as the diameter of the circle (smaller/larger circles depict less/more uncertainty associated with each outcome). In cases where less uncertainty is associated with outcomes (smaller blue circles), as potentially is the case with described options, the probability distribution inferred from those smaller, more discrete points would be sharper. Conversely, in cases where more uncertainty is associated with outcomes (larger, grey circles), as may be the case with experience options, the inferred probability distribution may be broader (because 'wider' data points in the distribution could overlap similar to kernel density estimation). We theorize this process could occur uniquely for each stimulus/choice option.

in outcome weighting. Thus, options with greater outcome uncertainty would be expected to yield a wider range of possible values, allowing for greater distortions in perceived probability in the sense that wider distributions provide higher/lower bounds for inferred values.

Notably, this model makes predictions consistent with recent literature on information-seeking which suggests that people prefer to find out information about positive outcomes more than negative outcomes [19,21,27]. From the perspective that the experience options were less informative (and thus more uncertain) than the description options, one could interpret our results in terms of participants wanting to resolve uncertainty about prospectively positive outcomes with greater upper bounds (and hence prefer the experience options when seeking gains) and would want to avoid those uncertain options with more extreme lower bounds when there is the prospect of loss. This could potentially explain why participants behaved as if the experience options were more likely than their equiprobable description counterparts. More generally, our results support the notion that outcome uncertainty related to a stimulus' 'informativeness' can influence probability perception and risk attitudes. Future work is required to integrate these concepts into a single model which learns both a distribution of probabilities and how much weight to place on positive and negative outcomes.

## 4.3. Influence of probability distortion on additional measures

Although we have focused on analysing choice patterns, additional support for the notion that outcome uncertainty modulates probability perception comes from our exploratory analysis of choice RTs. Participant choices were slower when choosing between experience stimuli than choosing between described options during both the training and main experimental blocks across both the gain and loss contexts. From the perspective of the probabilistic inference framework described above, these RT results suggest that more cognitive resources may be required to integrate over broader, less certain internal probability representations as compared to integrating over narrower, more certain internal probability representations. We also found that RTs were generally greater when participants mitigated losses as compared to sought gains. This RT result mirrors the choice pattern result that participant performance was generally lower when mitigating losses as compared to seeking gains. Taken together, this suggests that participants may have found the loss-version of our task more difficult than the gain-version. Consistent with this idea, we found that loss-context RTs during equiprobable trials increased as loss probabilities increased, potentially indicating that these trials engendered greater choice conflict. More generally, our RT analyses highlight that choice dynamics are also affected by probability distortion.

Finally, it is worth noting that participants' subjective reports of outcome likelihoods in the post-experiment survey (Exp 1, figure 6) were dramatically different from their real-choice patterns: they rated described options as more likely to occur than experience ones, and this size of the gap was largest for low-probability (20%) stimuli and trended towards inversion at high probability (80%). This description-oriented pattern fits with early reports of the DE gap [1,3]. The major similarity between our survey and early reports of the DE gap is that language was used to encourage participants to directly consider their inferences. To determine whether the experience of reflecting/considering mediates the difference between early description-oriented accounts of the DE gap and more recent experienced-oriented ones, it would be interesting to have participants report their inferences after the training blocks but before the main experimental blocks. One could then compare equiprobable choice patterns of participants reporting their inferences after training against participants who reported their inferences at the end of the experiment (as in this present dataset). It would be particularly interesting to see whether the post-training group's equiprobable choice patterns more closely matched their reported inferences than the post-experiment group. Taken together, the role of reflection/consideration on subjective inference remains unexplored and could prove useful for bridging the gap between early and recent accounts of the DE gap.

# 5. Conclusion

We assessed the DE gap as human participants made decisions between informative, described and uninformative, experience options where the difference between the probabilistic values of the choice options varies across trials. We directly assessed the effect of stimulus type/informativeness on inferred value through the notion of equiprobable gambles—trials where stimuli differ only in informativeness but have objectively equal probabilities to either win or lose points. We found that gain-seeking participants preferred experience stimuli across all tested probability levels (20%, 50%, and 80%) and, by contrast, loss-mitigating participants avoided the experience stimuli across all tested probability levels. Additional exploratory modelling with a distributional reinforcement learning model suggests that these choice patterns could have arisen from differential weightings of the outcomes associated with description and experience options. Our RT analyses indicated that participants required longer to respond when choosing between experience stimuli as compared to described options and were generally longer when mitigating losses as compared to seeking gains. Although we did not observe choice patterns consistent with early reports of the DE gap, our results are consistent with of risk-seeking for gains and risk-aversion when mitigating losses in serialized choice. Together, we interpret these findings as evidence that differential uncertainty associated with described and experience options may have distorted participants' perceptions of their associated probabilities, thereby altering their risk attitudes towards them.

Ethics. All procedures were consistent with the relevant German laws on human behavioural experimentation and were in accordance with the ethical standards of the National Research Committee and all guidelines described in the 1964 Declaration of Helsinki. All participants gave informed consent before testing and all data were anonymized.

Data accessibility. All raw data, stimuli, and MATLAB code used to analyse the data are available via the Open Science Framework and can be accessed from this link: https://osf.io/a8zf6/?view_only=b43e278fe15d4fe5a51dd52ee4bc9c30

Authors' contributions. T.W.E. conceived the idea. T.W.E., I.G.M. and V.M. designed the experiment. I.G.M. wrote the code to conduct the experiment and oversaw data collection. T.W.E. performed the analysis. T.W.E., I.G.M. and V.M. wrote the paper.

Competing interests. We declare we have no competing interests

Funding. T.W.E. was supported by an Alexander von Humboldt Postdoctoral Fellowship. Publication charges were supported by the Open Access Publishing Fund of University of Tübingen.

Acknowledgements. We thank Katharina Brecht for early discussions during the design of the experiment.

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
