## [Peer Review File · Royal Society Open Science]

Review History

RSOS-210307.R0 (Original submission)

Review form: Reviewer 1

Is the manuscript scientifically sound in its present form?

No

Are the interpretations and conclusions justified by the results?

No

Is the language acceptable?

Yes

Do you have any ethical concerns with this paper?

No

Have you any concerns about statistical analyses in this paper?

Yes

Recommendation?

Major revision is needed (please make suggestions in comments)

Comments to the Author(s)

This manuscript assesses the Description-Experience (DE) gap in a decision-making task where participants have to choose between descriptive stimuli (filled bars depicting 20%, 50%, or 80% outcome probability) and experience stimuli (3 different emoji pictures associated with the same 3 outcome probabilities). One group of participants (N=30, gain group) played to gain points and another group (N=30, loss group) played to avoid losing points. Participants in the gain group exhibited a preference for experience over descriptive stimuli, while participants in the loss group showed the opposite pattern. In both groups, the DE gap was maximal for intermediate, more uncertain, probabilities (50%), a finding at odds with some of the past literature.

Overall, this is an interesting and well-written manuscript, with the potential of contributing some novel insights on the factors influencing the DE gap. However, I have several concerns, some of which I hope can be addressed by the authors.

Major concerns:

1) My main concern is that while the authors do find a strong relationship between initial training performance and the size of the equiprobable bias (Figure 3), they do not do any analyses to ensure that the patterns of DE gap they observe (and the effects of gain/loss, uncertainty, and their interaction) remain present when controlling for differences in initial learning. There are two analyses that I think are necessary here:

a. First, report the association between training performance and equiprobable bias separately for each probability condition. Does this give any insights into whether the higher DE gap observed at 50% probabilities could be driven by the fact that 50% probability is harder to learn than 20% or 80% during the training block?

b. Second, include training performance data as a covariate in the main analysis to ensure the effect of gain/loss context and uncertainty on experience choice remain when accounting for the effect of gain/loss context and uncertainty on training performance.

2) My second concern is the overall low sample size of the study, the between-subjects design, and the fact that the results are not replicated. With the overall replication crisis the field is currently facing, I believe it would be extremely valuable, if possible for the authors, to replicate these results in an independent sample, while at the same time potentially directly addressing some of the concerns/limitations the authors report in the current version of the task (e.g. using emojis with potential meaning, not measuring perceived probabilities after training, etc). In addition, I believe it would be extremely valuable to test whether the difference between the gain and loss groups persists in a within-subjects design. Because the current group sizes are relatively small (and no power analysis or effect sizes are reported), it is possible that some of the group comparisons are underpowered and that other differences between groups could explain the gain/loss effects. For example, we know that economic biases such as risk aversion, loss aversion, ambiguity aversion, probability weighting, etc, are highly variable across people, and since those were not measured, we do not know if they could be causing some of the differences between groups. I do understand, however, that running replication studies is not always possible, in which case these concerns should be acknowledged and discussed in depth in the discussion.

3) My third concern is more of a suggestion. I believe the inference process the authors describe in the discussion and in Figure 6 is interesting but remain quite vague. Would there be any possibility to actually test this model empirically?

a. One option could be to build a generative model using different probability distributions depending on the assumed uncertainty and inferred probability, generate data using that model to "play" the task, both in the loss and gain domains, then see if the patterns of DE gap can be

recovered from this model-generated data. This would be a first, very useful step in validating the model.

b. If a. works, why not try to fit the model to each subject's data and actually estimate what the mean and the width of the distribution is for each stimulus? Assuming each distribution could be modelled with a beta distribution (<https://distribution-explorer.github.io/continuous/beta.html>), this means that 2 parameters would need to be estimated for each stimulus, which may be a lot. But maybe some simplification could be made (e.g. assuming the width of the distribution is different between D and E stimuli, but the same within each category).

4) I am worried about why training performance in the Descriptive block is so low, especially in the loss group (x-axis for Figure 3A), with ~10 subjects showing a performance of <0.6. Surely, performance in the Descriptive training block (and the pure Descriptive trials in the main task) should be at ceiling? Why would participants ever make any mistake, except for inattention, in these trials? Could it be that participants did not understand what the descriptive stimuli mean?

5) The introduction has a lot of "method-like" text and lacks a bit more background and rationale about why this work is important. The authors do point out some gaps/problems in previous studies, but do not really make the point of why it is important to address them. It would also be great if the introduction could touch on the risk vs uncertainty literature (mentioned in the discussion, but highly relevant to the study), and maybe as well on the relevance of risk vs ambiguity aversion.

6) Recent literature on information-seeking (e.g. Charpentier et al, 2018, PNAS; Sharot & Sunstein, 2020, Nature Human Behaviour; van Lieshout et al, 2020, Current Opinion in Behavioral Sciences) suggests that people prefer to find out information about positive outcomes more than about negative outcomes. Could this bias play a role in the current study? One could imagine that participants want to resolve uncertainty (hence the preference for experience over descriptive stimuli) more in the gain than loss domain.

Minor comments (in order of the text):

- In the abstract, the effect of uncertainty (DE gap maximal for 50% probabilities) is not clearly conveyed.
- In general throughout the text, I would avoid calling the gain/loss manipulation "risk context". Maybe "context" on its own is enough, or "outcome valence", is better. Having the word "risk" is confusing as it makes it sound like it's about the explicit probabilities.
- Introduction, p.5: it's unclear what "an S-shaped stimulus x probability profile" means. If this is the authors' predicted pattern of results, maybe they could add a panel to Figure 1 to illustrate it?
- Stimuli, p.7: were participants explicitly told about the probabilities associated with the rectangles in the descriptive stimuli? If so, then this makes point 4) above, as well as the perceived probabilities plotted in Figure 5 for the Descriptive condition, even more worrying. If subjects were not explicitly told the probabilities, please clarify what they were told exactly. If they were not told much, could this mean the descriptive stimuli are in fact more akin to experience stimuli?
- Procedure, p.7: were subjects only compensated with course credit? If the task was not incentive-compatible (i.e. outcomes have no impact on final payment), could this explain some of the differences with previous studies in the observed effects. If outcomes do not matter, this could make people more motivated to resolve uncertainties ("intrinsic" reward of gaining knowledge) than to maximize outcomes, for example.
- Results: please report effect sizes throughout.

- Was there any learning effect during the main task? For example was a win more likely to lead to participants picking that stimulus again later or a loss related to avoiding that stimulus later?
- Discussion, p.23: classic prospect theory findings suggest risk aversion for gains and risk-seeking in the loss domain (a direct consequence of the value function being concave for gains and convex for losses), so the authors' claim that "decision makers are risk-seeking when there is the prospect of gain but are risk-averse with the prospect of a loss" is wrong. In addition, the reference Yamada et al (2013) does not seem appropriate here. This is a monkey study in which only rewards were used and in which monkeys were found to be mildly risk averse for gains, similar to humans. There are a couple of other instances in the discussion where the authors mention how their results are consistent with this pattern of risk seeking for gains for risk aversion for losses (e.g., p.28). Please alter those accordingly, since this interpretation is actually at odds with the literature.
- What do the circles mean in Figure 6?
- Code sharing: thank you for sharing the raw data and the stimuli; however, it doesn't seem like the analysis code was shared.

Review form: Reviewer 2 (Jack Stecher)

Is the manuscript scientifically sound in its present form?

Yes

Are the interpretations and conclusions justified by the results?

No

Is the language acceptable?

Yes

Do you have any ethical concerns with this paper?

No

Have you any concerns about statistical analyses in this paper?

Yes

Recommendation?

Reject

Comments to the Author(s)

See attached letter (Appendix A).

Decision letter (RSOS-210307.R0)

Dear Dr Elston

The Editors assigned to your paper RSOS-210307 "Outcome uncertainty influences probability perception and risk attitudes" have now received comments from reviewers and would like you to revise the paper in accordance with the reviewer comments and any comments from the Editors. Please note this decision does not guarantee eventual acceptance.

Please submit your revised manuscript and required files (see below) no later than 21 days from today's (ie 04-Jun-2021) date. Note: the ScholarOne system will 'lock' if submission of the revision is attempted 21 or more days after the deadline. If you do not think you will be able to meet this deadline please contact the editorial office immediately.

on behalf of Dr Oliver Robinson (Associate Editor) and Essi Viding (Subject Editor)
openscience@royalsociety.org

Associate Editor Comments to Author (Dr Oliver Robinson):

Comments to the Author:

The reviewers make some very clear recommendations to improve the manuscript. I am happy to consider a revision, but can you:

- 1) Replicate the study (since this is a rapid online study)? Or provide a clear rationale for why this is not possible.
- 2) Discuss and report power/effect sizes?
- 3) Ensure that the baseline effects are clearly accounted for in the current data ?
- 4) Discuss and include the relevant literature cited by the reviewers and discuss how it relates to this paper?
- 5) ensure the analysis code is shared?

Reviewer comments to Author:

Reviewer: 1

Comments to the Author(s)

This manuscript assesses the Description-Experience (DE) gap in a decision-making task where participants have to choose between descriptive stimuli (filled bars depicting 20%, 50%, or 80% outcome probability) and experience stimuli (3 different emoji pictures associated with the same 3 outcome probabilities). One group of participants (N=30, gain group) played to gain points and another group (N=30, loss group) played to avoid losing points. Participants in the gain group exhibited a preference for experience over descriptive stimuli, while participants in the loss group showed the opposite pattern. In both groups, the DE gap was maximal for intermediate, more uncertain, probabilities (50%), a finding at odds with some of the past literature.

Overall, this is an interesting and well-written manuscript, with the potential of contributing some novel insights on the factors influencing the DE gap. However, I have several concerns, some of which I hope can be addressed by the authors.

Major concerns:

1) My main concern is that while the authors do find a strong relationship between initial training performance and the size of the equiprobable bias (Figure 3), they do not do any analyses to ensure that the patterns of DE gap they observe (and the effects of gain/loss, uncertainty, and their interaction) remain present when controlling for differences in initial learning. There are two analyses that I think are necessary here:

a. First, report the association between training performance and equiprobable bias separately for each probability condition. Does this give any insights into whether the higher DE gap observed at 50% probabilities could be driven by the fact that 50% probability is harder to learn than 20% or 80% during the training block?

b. Second, include training performance data as a covariate in the main analysis to ensure the effect of gain/loss context and uncertainty on experience choice remain when accounting for the effect of gain/loss context and uncertainty on training performance.

2) My second concern is the overall low sample size of the study, the between-subjects design, and the fact that the results are not replicated. With the overall replication crisis the field is currently facing, I believe it would be extremely valuable, if possible for the authors, to replicate these results in an independent sample, while at the same time potentially directly addressing some of the concerns/limitations the authors report in the current version of the task (e.g. using emojis with potential meaning, not measuring perceived probabilities after training, etc). In addition, I believe it would be extremely valuable to test whether the difference between the gain and loss groups persists in a within-subjects design. Because the current group sizes are relatively small (and no power analysis or effect sizes are reported), it is possible that some of the group comparisons are underpowered and that other differences between groups could explain the gain/loss effects. For example, we know that economic biases such as risk aversion, loss aversion, ambiguity aversion, probability weighting, etc, are highly variable across people, and since those were not measured, we do not know if they could be causing some of the differences between groups. I do understand, however, that running replication studies is not always possible, in which case these concerns should be acknowledged and discussed in depth in the discussion.

3) My third concern is more of a suggestion. I believe the inference process the authors describe in the discussion and in Figure 6 is interesting but remain quite vague. Would there be any possibility to actually test this model empirically?

a. One option could be to build a generative model using different probability distributions depending on the assumed uncertainty and inferred probability, generate data using that model to "play" the task, both in the loss and gain domains, then see if the patterns of DE gap can be

recovered from this model-generated data. This would be a first, very useful step in validating the model.

b. If a. works, why not try to fit the model to each subject's data and actually estimate what the mean and the width of the distribution is for each stimulus? Assuming each distribution could be modelled with a beta distribution (<https://distribution-explorer.github.io/continuous/beta.html>), this means that 2 parameters would need to be estimated for each stimulus, which may be a lot. But maybe some simplification could be made (e.g. assuming the width of the distribution is different between D and E stimuli, but the same within each category).

4) I am worried about why training performance in the Descriptive block is so low, especially in the loss group (x-axis for Figure 3A), with ~10 subjects showing a performance of <0.6. Surely, performance in the Descriptive training block (and the pure Descriptive trials in the main task) should be at ceiling? Why would participants ever make any mistake, except for inattention, in these trials? Could it be that participants did not understand what the descriptive stimuli mean?

5) The introduction has a lot of "method-like" text and lacks a bit more background and rationale about why this work is important. The authors do point out some gaps/problems in previous studies, but do not really make the point of why it is important to address them. It would also be great if the introduction could touch on the risk vs uncertainty literature (mentioned in the discussion, but highly relevant to the study), and maybe as well on the relevance of risk vs ambiguity aversion.

6) Recent literature on information-seeking (e.g. Charpentier et al, 2018, PNAS; Sharot & Sunstein, 2020, Nature Human Behaviour; van Lieshout et al, 2020, Current Opinion in Behavioral Sciences) suggests that people prefer to find out information about positive outcomes more than about negative outcomes. Could this bias play a role in the current study? One could imagine that participants want to resolve uncertainty (hence the preference for experience over descriptive stimuli) more in the gain than loss domain.

Minor comments (in order of the text):

- In the abstract, the effect of uncertainty (DE gap maximal for 50% probabilities) is not clearly conveyed.

- In general throughout the text, I would avoid calling the gain/loss manipulation "risk context". Maybe "context" on its own is enough, or "outcome valence", is better. Having the word "risk" is confusing as it makes it sound like it's about the explicit probabilities.

- Introduction, p.5: it's unclear what "an S-shaped stimulus x probability profile" means. If this is the authors' predicted pattern of results, maybe they could add a panel to Figure 1 to illustrate it?

- Stimuli, p.7: were participants explicitly told about the probabilities associated with the rectangles in the descriptive stimuli? If so, then this makes point 4) above, as well as the perceived probabilities plotted in Figure 5 for the Descriptive condition, even more worrying. If subjects were not explicitly told the probabilities, please clarify what they were told exactly. If they were not told much, could this mean the descriptive stimuli are in fact more akin to experience stimuli?

- Procedure, p.7: were subjects only compensated with course credit? If the task was not incentive-compatible (i.e. outcomes have no impact on final payment), could this explain some of the differences with previous studies in the observed effects. If outcomes do not matter, this could make people more motivated to resolve uncertainties ("intrinsic" reward of gaining knowledge) than to maximize outcomes, for example.

- Results: please report effect sizes throughout.

- Was there any learning effect during the main task? For example was a win more likely to lead to participants picking that stimulus again later or a loss related to avoiding that stimulus later?

- Discussion, p.23: classic prospect theory findings suggest risk aversion for gains and risk-seeking in the loss domain (a direct consequence of the value function being concave for gains and convex for losses), so the authors' claim that "decision makers are risk-seeking when there is the prospect of gain but are risk-averse with the prospect of a loss" is wrong. In addition, the reference Yamada et al (2013) does not seem appropriate here. This is a monkey study in which only rewards were used and in which monkeys were found to be mildly risk averse for gains, similar to humans. There are a couple of other instances in the discussion where the authors mention how their results are consistent with this pattern of risk seeking for gains for risk aversion for losses (e.g., p.28). Please alter those accordingly, since this interpretation is actually at odds with the literature.

- What do the circles mean in Figure 6?

- Code sharing: thank you for sharing the raw data and the stimuli; however, it doesn't seem like the analysis code was shared.

Reviewer: 2

Comments to the Author(s)

See attached letter.

===PREPARING YOUR MANUSCRIPT===

===PREPARING YOUR REVISION IN SCHOLARONE===

To revise your manuscript, log into <https://mc.manuscriptcentral.com/rsos> and enter your Author Centre - this may be accessed by clicking on "Author" in the dark toolbar at the top of the

page (just below the journal name). You will find your manuscript listed under "Manuscripts with Decisions". Under "Actions", click on "Create a Revision".

Author's Response to Decision Letter for (RSOS-210307.R0)

See Appendix B.

Decision letter (RSOS-210307.R1)

Dear Dr Elston,

It is a pleasure to accept your manuscript entitled "Outcome uncertainty influences probability perception and risk attitudes" in its current form for publication in Royal Society Open Science. The comments of the reviewer(s) who reviewed your manuscript are included at the foot of this letter.

Please ensure that you send to the editorial office an editable version of your accepted manuscript, and individual files for each figure and table included in your manuscript. You can send these in a zip folder if more convenient. Failure to provide these files may delay the processing of your proof.

on behalf of Dr Oliver Robinson (Associate Editor) and Essi Viding (Subject Editor)
openscience@royalsociety.org

Associate Editor Comments to Author (Dr Oliver Robinson):
Associate Editor

Comments to the Author:

The authors are to be commended for a thorough and comprehensive revision, with particular commendation for completing a fully pre-registered replication study. I think the manuscript is substantially improved by addressing the reviewers comments and am delighted to recommend it for acceptance at RSOS.

Appendix A

Review of “Outcome uncertainty influences probability perception and risk attitudes”

This manuscript explores attitudes toward risk in hypothetical decisions (only points are at stake, not participant payments), focusing specifically on differences in attitudes toward gains versus losses and the dependence of these attitudes on whether participants learn probabilities from a description or from a sampling task. Abdellaoui et al. (2011) address a similar research question (although they look at both hypothetical gambles and decisions with real incentives). They report a large effect of descriptions versus experience in gambles involving gains, but little effect in gambles involving losses. I would urge the authors of the current manuscript to highlight their incremental contribution compared with that of Abdellaoui et al..

The reason I draw attention to the use of purely hypothetical incentives is that there is considerable debate over whether reported decisions without incentives reflect how participants behave, or whether it reflects how they imagine themselves. In many contexts, there is little difference, and hypothetical choice can be a suitable design. However, in decision making under risk, there are known substantial differences. The literature is large, and I will cite only a few examples. In an early study, Feather (1959) demonstrates that participants are more risk averse when making hypothetical decisions; Battalio et al. (1990) demonstrates this result hold for both gains and losses,

and Holt and Laury (2002) shows that the difference (they focus only on gains) becomes substantial as stakes increase. In the context of the current study, the design would amplify how risk seeking participants are with losses (if we follow the predictions of prospect theory) and would dampen the level of risk aversion for gains. Thus, using only hypothetical payments would create a systematic difference for reasons unrelated to the current research focus. Somewhat related, Charness et al. (2010) finds that a fixed payment without incentives leads to more judgment errors than an incentive payment (their study paid \$2 for participating in one treatment and \$4 for a correct answer in another).

The analysis of the post-experiment survey, while not central to the main argument, needs much greater rigor if it is to be included. For instance, on p. 27, the manuscript states that participants “rated described options as more likely to occur than experience ones and the size of this gap was largest for low-probability (20%) stimuli and tended toward inversion at high probability (80%).” The difficulty with this statement is that the stimuli were all Binomial(n, p) lotteries. As first noted in Fox and Hadar (2006), this creates a confound due to the skewness of the binomial distribution, which is $(1 - 2p)/\sqrt{np(1 - p)}$. For $p < 1/2$, the skewness is positive: a small number of participants would be expected to observe more than np successes, and a large number would be expected to observe fewer. As p increases, this would diminish and tend toward inversion. A between-participants analysis is clearly inappropriate.

To correct for this, I suggest the authors conduct a within-participants analysis. For each participant, take the sample that the participant drew on the experience options. Compare the fraction of successes to the corresponding described probability. In the cases in which the sample exceeds the described probability a of 20%, did the participants still treat the described probability as more likely? Then do the same analysis for described probabilities of 50% and 80% and the corresponding samples on an individual level for the de-

scription options. Is there a systematic variation with the success probability within participants?

The discussion of risk versus uncertainty is at odds with conventional usage. Since Knight (1921), *risk* has been used to characterize settings with known probabilities, and *uncertainty* has generally been used to characterize settings without known probabilities. It is strange to claim that risk refers to “situations where outcomes are deterministic” (what exactly is at risk then?). Similarly, uncertainty does not conventionally have anything to do with learning, inference, or repeated experience. A decision maker can face a situation with basis for assigning any definite probability to any outcome (a point raised in Kolmogorov, 1983). Repetition has nothing to do with this.

Some of the other references to the literature need more careful statements. For instance, the manuscript reads as if Kahneman and Tversky (1979) study probabilities that are inferred through experience (their focus is elsewhere, specifically on questioning the appropriateness of expected utility theory and drawing attention to the idea probability weighting function). A better early reference is the study by Davidson et al. (1957), in which participants are given a specially manufactured die with nonsense syllables on each face. The participants are allowed to practice with the die, in order to give them the experience of equally probable outcomes. The manuscript also states that most prior work gives a choice between an experience option and a described certain option. This is not the case in Hertwig et al. (2004), which the manuscript cites.

As a final suggestion, I recommend that the authors try to state the research question more explicitly. The current writing discusses various comparisons, but does not state in a straightforward way what the study aims to test and what the broader significance of the question is. The discussion of response times comes across as an aside, and the reasons for excluding fast response times (which might include participants who have a solid understanding of

the task) and slow ones (which may include those who are learning) needs more justification and a better connection to the main research question. I understand that fast and slow respondents may be thought of as a source of noise due to lack of engagement, but it would be helpful to have some discussion and perhaps some reassurance that including the entire sample does not drastically affect the significance or direction of the findings.

References

- M. Abdellaoui, O. L’Haridon, and C. Paraschiv. Experienced vs. described uncertainty: Do we need two prospect theory specifications? *Management Science*, 57(10):1879–1895, 2011.
- R. C. Battalio, J. H. Kagel, and K. Jiranyakul. Testing between alternative models of choice under uncertainty: Some initial results. *Journal of Risk and Uncertainty*, 3(1):25–50, 1990.
- G. Charness, E. Karni, and D. Levin. On the conjunction fallacy in probability judgment: New experimental evidence regarding Linda. *Games and Economic Behavior*, 68(2):551–556, 2010.
- D. Davidson, P. Suppes, and S. Siegel. *Decision Making: An Experimental Approach*. Stanford University Press, 1957.
- N. T. Feather. Subjective probability and decision under uncertainty. *Psychological Review*, 66(3):150–164, 1959.
- C. R. Fox and L. Hadar. “Decisions from experience” = sampling error + prospect theory: Reconsidering Hertwig, Barron, Weber & Erev. *Judgment and Decision Making*, 1(2):159–161, 2006.
- R. Hertwig, G. Barron, E. U. Weber, and I. Erev. Decisions from experience and the effect of rare events in risky choice. *Psychological Science*, 15(8): 534–539, 2004.

- C. A. Holt and S. K. Laury. Risk aversion and incentive effects. *American Economic Review*, 92(5):1644–1655, 2002.
- D. Kahneman and A. Tversky. Prospect theory: An analysis of decision under risk. *Econometrica*, 47(2):263–292, 1979.
- F. H. Knight. *Risk, Uncertainty, and Profit*. Hart, Schaffner & Marx; Houghton Mifflin Company, Boston, 1921.
- A. N. Kolmogorov. *On Logical Foundations of Probability Theory*, volume 1021 of *Lecture Notes in Mathematics*, pages 1–5. Springer-Verlag, Berlin Germany, 1983.

Appendix B

Dear Dr. Robinson,

We would now like to submit the revised version of manuscript RSOS-210307 "Outcome uncertainty influences probability perception and risk attitudes". We thank you and the reviewers for the many helpful comments, to which we have responded as described below. To improve the readability of this letter, the reviewers' comments are in bold-face and our responses are not.

We hope the revised version is acceptable for publication. For your convenience and that of the reviewers, we have indicated the most significant changes in the revision by highlighting them in **blue font**. Thank you for your continued consideration of the manuscript.

Sincerely,

Thomas Elston & co-authors

Comments of the Editor

The reviewers make some very clear recommendations to improve the manuscript.

We agree that the reviewers make some excellent suggestion to further strengthen our manuscript. As you can see, we have tried to address all of the reviewers' suggestions, as indicated below in our point-by-point responses to them.

I am happy to consider a revision, but can you:

Thank you for these highlights. As can be seen next, we have also addressed all of your specific suggestions in our revised manuscript.

1) Replicate the study (since this is a rapid online study)? Or provide a clear rationale for why this is not possible.

We have conducted another (preregistered) experiment ($N = 60$). As can be seen in our revised manuscript, the results of this new experiment ("Experiment 2") closely resemble the ones found for the initial experiment ("Experiment 1") and are fully incorporated into our revised manuscript.

2) Discuss and report power/effect sizes?

We have added the effect sizes for the results of the two experiments. Furthermore, we have added a subheading in our participants section (page 6) in which we state that a power analysis to detect a medium sized effect ($d = .50$) between "equal description vs. experience" trials within one context condition (e.g., 20% filled bar and emoji with a 20% gain probability) would have suggested that we have over 80% (Experiment 1) and over 99% (Experiment 2) power to detect a significant effect (one-sided) with a significance level of 5%.

3) Ensure that the baseline effects are clearly accounted for in the current data?

As mentioned above, we have replicated the results using a within-subject design and as such it seems very unlikely to us that the differences between the two groups of participants could explain the gain/loss effects (see also our response to the second comment of Reviewer 1). Furthermore, as can be seen in our specific response to the first comments of Reviewer 1, we have also explored for both Experiment 1 and 2 the extent to which differences in initial learning drive the pattern observed in the main experimental blocks. As mentioned in a in our revised results section on page 18, these additional analyses demonstrated that the key probability*context effect in the main block equiprobable choice conditions remain present when controlling for learning effects.

4) Discuss and include the relevant literature cited by the reviewers and discuss how it relates to this paper?

Yes, the proposed literature fits well to the current study and we have covered this literature in our introduction and/or discussion (highlighted in blue font in the revised manuscript). Please see our specific responses to the reviewers.

5) ensure the analysis code is shared?

We have uploaded the MATLAB code used for the experiment to our Open Science Framework page and have directly mentioned this link under the Data Availability subheading on page 5 of the revised manuscript. Here is that link:

https://osf.io/a8zf6/?view_only=b43e278fe15d4fe5a51dd52ee4bc9c30

Comments of Reviewer 1

Overall, this is an interesting and well-written manuscript, with the potential of contributing some novel insights on the factors influencing the DE gap. However, I have several concerns, some of which I hope can be addressed by the authors.

Thank you for these encouraging words and for the constructive comments. As is described in more detail below, we have addressed all of your comments and think they substantially strengthened our paper.

1) My main concern is that while the authors do find a strong relationship between initial training performance and the size of the equiprobable bias (Figure 3), they do not do any analyses to ensure that the patterns of DE gap they observe (and the effects of gain/loss, uncertainty, and their interaction) remain present when controlling for differences in initial learning. There are two analyses that I think are necessary here:

Although the additional analyses below do not seem to suggest that the pattern of the experimental blocks depends solely on learning, we would also like to note that we think our results would be also interesting if this would not have been the case since our novel experimental approach was in general to see whether a DE gap is observed when there are no “sure bet” option/no intermediate computation of probabilities. Beyond this rather general point, however, we agree that the two suggested analyses (and corresponding results) are helpful to better understand the reported effects.

a. First, report the association between training performance and equiprobable bias separately for each probability condition. Does this give any insights into whether the higher DE gap observed at 50% probabilities could be driven by the fact that 50% probability is harder to learn than 20% or 80% during the training block?

Thank you for this thoughtful suggestion. We have implemented this analysis (presented in Figure S3 and precisely detailed in the Supplemental Data). It is mentioned on page 18 of the revised manuscript. We found no evidence that the DE gaps observed at different probabilities were related to differences in training performance at different probabilities.

b. Second, include training performance data as a covariate in the main analysis to ensure the effect of gain/loss context and uncertainty on experience choice remain when accounting for the effect of gain/loss context and uncertainty on training performance.

This is also a good suggestion and we performed the proposed analyses for both Experiment 1 and Experiment 2. While training performance was indeed a significant covariate, the key Probability*Context effect explained a significant proportion of variance beyond that of training performance. We have mentioned this analysis in the results section on Page 18 of the revised manuscript.

2) My second concern is the overall low sample size of the study, the between-subjects design, and the fact that the results are not replicated. With the overall replication crisis the field is currently facing, I believe it would be extremely valuable, if possible for the authors, to replicate these results in an independent sample, while at the same time potentially directly addressing some of the concerns/limitations the authors report in the current version of the task (e.g. using emojis with potential meaning, not measuring perceived probabilities after training, etc). In addition, I believe it would be extremely valuable to test whether the difference between the gain and loss groups persists in a within-subjects design. Because the current group sizes are relatively small (and no power analysis or effect sizes are reported), it is possible that some of the group comparisons are underpowered and that other differences between groups could explain the gain/loss effects. For example, we know that economic biases such as risk aversion, loss aversion, ambiguity aversion, probability weighting, etc, are highly variable across people, and since those were not measured, we do not know if they could be causing some of the differences between groups. I do understand, however, that running replication studies is not always possible, in which case these concerns should be acknowledged and discussed in depth in the discussion.

We appreciate your concern here and would also like to emphasize that our initial study was at least sufficiently powered to detect a medium effect between experience and description stimuli in equiprobable trials within each context (see the sample size justification on page 6). However, we agree that it would be valuable to investigate whether we could shift individual's DE gaps across gain and loss contexts within a single experimental session. Therefore, we decided to see whether the findings of our initial experiment (Exp 1) would replicate when using a within-subject design and a large sample size ($N = 60$; see also our response to your minor comment 5 in this regard). It should be also emphasized that we implemented some real-word incentives in this new experiment to empirically address a comment by Reviewer #2 who wondered whether our subjects would choose differently were they working for some kind of real-world incentive. As can be seen in our revised result section, all key effects were replicated in this within-subject design, meaning the equiprobable biases we observed in the between-subject design could be obtained (and shifted back and forth) in the same participant within a single ~40 minute experimental session. Thus, we consider it as highly unlikely that group differences could explain the findings observed in our first experiment.

Please also note that we decided not to add any subjective questionnaires to this new experiment in order to avoid inducing additional strategies in participants choice behavior

after asking about the probabilities (which is obviously problematic in the within-subject design).

3) My third concern is more of a suggestion. I believe the inference process the authors describe in the discussion and in Figure 6 is interesting but remain quite vague. Would there be any possibility to actually test this model empirically?

a. One option could be to build a generative model using different probability distributions depending on the assumed uncertainty and inferred probability, generate data using that model to "play" the task, both in the loss and gain domains, then see if the patterns of DE gap can be recovered from this model-generated data. This would be a first, very useful step in validating the model.

b. If a. works, why not try to fit the model to each subject's data and actually estimate what the mean and the width of the distribution is for each stimulus? Assuming each distribution could be modelled with a beta distribution (<https://distribution-explorer.github.io/continuous/beta.html>), this means that 2 parameters would need to be estimated for each stimulus, which may be a lot. But maybe some simplification could be made (e.g. assuming the width of the distribution is different between D and E stimuli, but the same within each category).

We appreciate your interest in the inference concept we develop in the discussion and hope that it provides a useful first step how the underlying processes giving rise to the –in our view interesting– empirical finding could be conceptualized. That said, we agree that it is underspecified and can see how/why the beta distribution could be an excellent tool for implementing the differential impact of outcomes on likelihoods within such a model. We are now working on such a model (a Bayesian agent which learns probabilities with differential uncertainty with outcomes modelled as beta distributions). This work is still in progress and would be better presented in a separate, model-focused publication where several model classes could be directly compared.

Nevertheless, we appreciate your comment that the case for our inference model concept could be strengthened by some formalism. For this paper, we've opted to use a distributional reinforcement learning (distRL) model, an emerging but established class of RL models which aims to learn a distribution of possible values associated with each choice option. The results of this model (detailed and presented in the Supplemental Data) suggest that our subjects differentially weighted the outcomes of experience and description stimuli in the gain and loss contexts, as predicted by our probabilistic inference concept. Interestingly, the parameters derived from the distRL model could predict individual subjects' DE gaps. Although the distRL model cannot explain *why* subjects would differentially weight choice outcomes as a function of stimulus type and context (as our probabilistic inference concept

might), we see these results as compatible with our theory and use them to further specify our probabilistic inference concept. Because this was primarily an exploratory analysis meant to offer another layer of characterization to our empirical findings, we present these results in the Supplemental Data (Figure S4) and only briefly refer to them in the revised manuscript in the revised General Discussion (see page 28).

4) I am worried about why training performance in the Descriptive block is so low, especially in the loss group (x-axis for Figure 3A), with ~10 subjects showing a performance of <0.6. Surely, performance in the Descriptive training block (and the pure Descriptive trials in the main task) should be at ceiling? Why would participants ever make any mistake, except for inattention, in these trials? Could it be that participants did not understand what the descriptive stimuli mean?

We appreciate this concern and would like to note that we obtained qualitatively similar results in both experiments when excluding participants based on their training performance (e.g. excluding when performance was less than 50% and/or 60%). However, because we were unsure of exactly where best to place that cutoff, we opted to retain all participants in the final analysis to avoid artificially inflating our results by cherry-picking which participants to include. Furthermore, several other factors like individual differences in inattention, boredom, curiosity (e.g., to see whether maybe the rules have changed) might have also contributed to the the pattern under question.

One reason why subjects would not be at ceiling performance when choosing between description options likely relates to the fact that we did not explicitly tell participants that the filledness of the bars corresponds to their outcome probabilities. Our fundamental concept with the description and experience stimuli was *informativeness* – that the features of description stimuli themselves provided a symbolic system which could differentially facilitate learning and performance as compared to the entirely uninformative emoji/experience stimuli. We would like to emphasize that subjects were much more accurate and responded more quickly when choosing between description options (as compared to experience options) in the training and main experiment blocks, suggesting that they used this increased informativeness to make their choices.

We appreciate how this might be puzzling for other readers and emphasized the notion of informativeness in both the Method section (pages 6 and 7) and General Discussion (page 25).

5) (i) The introduction has a lot of "method-like" text and lacks a bit more background and rationale about why this work is important. The authors do point out some gaps/problems in previous studies, but do not really make the point of why it is important to address them. (ii) It would also be great if the introduction could touch on the risk vs uncertainty literature (mentioned in the discussion, but highly relevant to the study), and maybe as well on the relevance of risk vs ambiguity aversion.

(i) Thank you for pointing this out – when we initially designed the study, we were heavily interested in these methodological aspects (since they distinguished our study from ones before). Please note however that the current study includes an important experimental design feature to better understand the underlying causes of probability perception. Furthermore, some methodological aspects are important to consider (and to mention) because they could directly affect prior conclusions related to the DE gap (e.g., sampling error). Thus, we felt that mentioning these methodological aspects which might affect theoretical conclusions is important. However, it is also clear from this comment that we need to do a better job at laying out why it is important to address these aspects in the present study. Therefore, we have removed the less relevant aspects of our our method when revising the introduction in order to more strongly highlight the critical equiprobable gamble concept. Furthermore, we have implemented several changes in structure and wording to emphasize more directly the need of the current research (highlighted in the revised introduction).

(ii) As mentioned in response to your earlier point, we were primarily motivated by the questions of the boundary conditions of the DE gap in equiprobable gambles. Therefore, to retain focus, we have only added an additional sentence to the introduction on this topic. However, we have expanded and clarified our treatment of the topic (along with including the information-seeking ideas mentioned in comment 6) in the General Discussion (see pages 27 and 28).

6) Recent literature on information-seeking (e.g. Charpentier et al, 2018, PNAS; Sharot & Sunstein, 2020, Nature Human Behaviour; van Lieshout et al, 2020, Current Opinion in Behavioral Sciences) suggests that people prefer to find out information about positive outcomes more than about negative outcomes. Could this bias play a role in the current study? One could imagine that participants want to resolve uncertainty (hence the preference for experience over descriptive stimuli) more in the gain than loss domain.

Thank you for these very interesting papers - we particularly enjoyed reading Charpentier et al. and can see why you suggested it! We agree that information-seeking effects could play a role in our subjects' behavior. To that end, we have incorporated these references throughout the paper and discussed an information-seeking interpretation directly on page 29 in the revised General Discussion. We had not considered this but can see the role it could play. Thank you again and we will certainly take this onboard as we think about future experiments.

Minor comments (in order of the text):

1. In the abstract, the effect of uncertainty (DE gap maximal for 50% probabilities) is not clearly conveyed.

Thank you for pointing this out. We have modified the abstract to specify that the maximal DE gap we observed occurred in the 50% probability condition.

2. In general throughout the text, I would avoid calling the gain/loss manipulation “risk context”. Maybe “context” on its own is enough, or “outcome valence”, is better. Having the word “risk” is confusing as it makes it sound like it’s about the explicit probabilities.

We apologize for the lack of clarity and we thank the reviewer for the specific suggestions: We have opted to use the term “context” in place of “risk context”.

3. Introduction, p.5: it’s unclear what “an S-shaped stimulus x probability profile” means. If this is the authors’ predicted pattern of results, maybe they could add a panel to Figure 1 to illustrate it?

Thank you for this good suggestion. We have opted to clarify what an *inverse-S-shaped stimulus x probability profile means in the text and added a panel, now Figure 1C, schematizing our hypothesis.

4. Stimuli, p.7: were participants explicitly told about the probabilities associated with the rectangles in the descriptive stimuli? If so, then this makes point 4) above, as well as the perceived probabilities plotted in Figure 5 for the Descriptive condition, even more worrying. If subjects were not explicitly told the probabilities, please clarify what they were told exactly. If they were not told much, could this mean the descriptive stimuli are in fact more akin to experience stimuli?

(Please see also our response to the major point 4)

Participants were not told in Experiment 1 (and also not in Experiment 2) that the description stimuli were explicit probability cues - they were only told to finish their session with the greatest number of points possible as we now more explicitly state in the revised method section. More important, as mentioned in our previous response and emphasized in the manuscript on page 25), the results show that these stimuli strongly influenced choice behavior. Specifically, participants generally performed better with the descriptive as compared to the experience stimuli in both the training and main, experimental blocks in both contexts and in both the within/between-subject designs. Together, these results support the notion that participants learned that the descriptive stimuli were more informative than the experience ones.

5. Procedure, p.7: were subjects only compensated with course credit? If the task was not incentive-compatible (i.e. outcomes have no impact on final payment), could this explain some of the differences with previous studies in the observed effects. If outcomes do not matter, this could make people more motivated to resolve uncertainties (“intrinsic” reward of gaining knowledge) than to maximize outcomes, for example.

Yes, in the between-subject Experiment 1 participants were only compensated with course credit. However, in our within-subject Experiment 2, the 10 participants (of of 60) with the

highest amount of accumulated points also obtained 10€-vouchers. Thus, the points in Experiment 2 were related to monetary incentives and the result pattern was qualitatively similar across experiments.

Of course, we cannot exclude if people would be more or less motivated with different outcomes (e.g., if every participants would have obtain money in proportion to their accumulated points). However, to our knowledge, there seems no general consensus what type/amount of incentives seems more or less appropriate to study decision-making under uncertainty and that the specific incentives varies in general across studies (see Reviewer 2's second point). However, we would like to emphasize that we did not see an effect of financial incentive across experiments. We briefly emphasize this issue in our revised General Discussion on page 24.

6. Results: please report effect sizes throughout.

We apologize for not including effect sizes in our initial submission. As you can see in our revised manuscript, we have added effect sizes to all statistical statements in the manuscript.

7. Was there any learning effect during the main task? For example, was a win more likely to lead to participants picking that stimulus again later or a loss related to avoiding that stimulus later?

Thank you for this interesting suggestion. We addressed this by doing both a win-stay and lose-shift analysis of choice patterns during the “pure description” and “pure experience” trials of the main, experimental blocks. Specifically, the win-stay analysis looked for all instances of when a choice of a particular image (e.g. 50% bar) yielded a win (+1 in the gain context; -0 in the loss context). Then, we found the nearest trials when that same image was presented again and asked how often participants selected that same option again. The same was done with the lose-shift lose-shift analysis: we found all instances where a choice of a particular image led to a loss (+0 in the gain context; -1 in the loss context) and then assessed how likely participants were to pick the alternative the next time the loss-yielding stimulus was presented (whatever the alternative may have been on that particular trial).

This analysis, presented in Figure S1 and described in detail in the Supplemental Data, did not reveal choice patterns consistent with ongoing learning. For example, we found that subjects were more likely to repeatedly choose a high-probability option regardless of whether that option yielded a positive outcome the last time it was picked. Similarly, subjects were less likely to repeatedly select a low-probability option regardless of it's outcome. Rather than a learning effect, these results suggest that participants formed stable beliefs about the option values and supports the notion that they understood the task.

We briefly refer to these results on page 15 of the revised manuscript.

8. Discussion, p.23: classic prospect theory findings suggest risk aversion for gains and risk-seeking in the loss domain (a direct consequence of the value function being concave for gains and convex for losses), so the authors' claim that "decision makers are risk-seeking when there is the prospect of gain but are risk-averse with the prospect of a loss" is wrong. In addition, the reference Yamada et al (2013) does not seem appropriate here. This is a monkey study in which only rewards were used and in which monkeys were found to be mildly risk averse for gains, similar to humans. There are a couple of other instances in the discussion where the authors mention how their results are consistent with this pattern of risk seeking for gains for risk aversion for losses (e.g., p.28). Please alter those accordingly, since this interpretation is actually at odds with the literature.

We apologize that our reference to prospect theory when describing risk-seeking was confusing. We initially wrote that sentence thinking of low probabilities, where prospect theory does predict risk seeking and where the DE gap was initially reported (Hertwig et al., 2004; Psych Science). We also note that Heilbrunner and Hayden's (2016, Psychonomic Bulletin & Review) paper that we based our study on also used the phrase „risk seeking for gains“ quite often for both motivating and interpreting their study.

However, we can see now that this is confusing since prospect theory differentially predicts risk-seeking for low probability options and risk-aversion for moderate and high-probability options. To avoid any confusion for readers of our revised manuscript, we have decided to remove explicit discussion of classic prospect theory along with the reference to Yamada et al. We have now rephrased the specific sentence/claim. Specifically, on page 26, we now state that the present findings seem to suggest that participants may be more risk-seeking for gains compared to losses when making serialized (repeated) choice (as opposed to single choices)—a finding which seems in line with previous studies (cf. Samuelson, 1963, *Scientia*; Gideon et al., 1987, *J. Exp. Psy. Learn. Mem. Cog.*).

We have also included information about the effects of serialized choice in both the introduction and a dedicated subheading in the General Discussion on page 26.

- What do the circles mean in Figure 6?

We apologize for the lack of clarity here. The size of the circles is meant to highlight different amounts of uncertainty associated with the outcomes of choices from description and experience (each circle denoting one outcome); larger circles indicate more uncertain outcomes. We have modified the phrase on page 28 in the General Discussion and made direct mention of the circle sizes in both the Figure itself (now Figure 7) and its legend for clarity.

- Code sharing: thank you for sharing the raw data and the stimuli; however, it doesn't seem like the analysis code was shared.

We have uploaded the MATLAB code used to analyze the data to our Open Science Framework page at the following link:

https://osf.io/a8zf6/?view_only=b43e278fe15d4fe5a51dd52ee4bc9c30

Comments of Reviewer 2

This manuscript explores attitudes toward risk in hypothetical decisions (only points are at stake, not participant payments), focusing specifically on differences in attitudes toward gains versus losses and the dependence of these attitudes on whether participants learn probabilities from a description or from a sampling task. Abdellaoui et al. (2011) address a similar research question (although they look at both hypothetical gambles and decisions with real incentives). They report a large effect of descriptions versus experience in gambles involving gains, but little effect in gambles involving losses. I would urge the authors of the current manuscript to highlight their incremental contribution compared with that of Abdellaoui et al.

Thank you for pointing out this study to us. It is true that –similar to the present study– Abdellaoui et al. investigated description- and experience-based decision as a function of gain/loss-contexts. However, the goal of the present study and the corresponding experimental design to tackle this goal differs in important ways from Abdellaoui et al.'s. As we mentioned in our introduction, we explicitly designed our study to assess whether the DE gap could be directly observed without any intermediate step, such as computing a certainty equivalent, which seems necessary when including sure-bet option as done in Abdellaoui et al. However, we can see that we should have been clearer about how this specific methodological aspects relates to the goal of the present study and helps to advance our theoretical understanding of probabilistic decision making under different contexts—as we also acknowledge in our responses to another comment of this reviewer („As a final suggestion, I recommend that the authors try to state the research question more explicitly..“) and Reviewer 1 (see major point 5). We have emphasized these methodological differences in the revised General Discussion (see page 26).

It is also worth highlighting that Abdellaoui et al.'s results were substantially different from ours in that they found evidence of a description-oriented DE gap most prominent at low probabilities and we found an experience-oriented DE gap most prominent at 50%. Because many of the factors present in Abdellaoui et al.'s study were also present in ours it is worth considering why our results are so different from theirs. We address this study in depth, dedicating more than a full paragraph to it in the General Discussion (see page 26) and explain the differences between our study and Abdellaoui et al.'s in terms of contextual factors known to influence risky choice that were different between our two studies to offer some thoughts about why studies like Abdellaoui et al.'s tend to find a description-oriented DE gap whereas studies like ours tend to find an experience-oriented DE gap.

The reason I draw attention to the use of purely hypothetical incentives is that there is considerable debate over whether reported decisions without incentives reflect how participants behave, or whether it reflects how they imagine themselves. In many contexts, there is little difference, and hypothetical choice can be a suitable design.

However, in decision making under risk, there are known substantial differences. The literature is large, and I will cite only a few examples. In an early study, Feather (1959) demonstrates that participants are more risk averse when making hypothetical decisions; Battalio et al. (1990) demonstrates this result hold for both gains and losses, and Holt and Laury (2002) shows that the difference (they focus only on gains) becomes substantial as stakes increase. In the context of the current study, the design would amplify how risk seeking participants are with losses (if we follow the predictions of prospect theory) and would dampen the level of risk aversion for gains. Thus, using only hypothetical payments would create a systematic difference for reasons unrelated to the current research focus. Somewhat related, Charness et al. (2010) finds that a fixed payment without incentives leads to more judgment errors than an incentive payment (their study paid \$2 for participating in one treatment and \$4 for a correct answer in another).

(Reviewer 1, minor point 5, raised a similar point—please also see our specific response there)

To begin with, we would first like to thank the reviewer for this comment and for all the references. Admittedly, we were not aware about this literature, but we can see now that the results could have been potentially different when using real-word incentives instead of just relying on hypothetical points (as we did in Experiment 1). Thus, we have decided to investigate whether the results would replicate in a new experiment (Experiment 2) in which participants were motivated by real-word incentives—that is, the 10 (out of 60) participants with the highest amount of points would additionally obtain 10€-vouchers for participation. As reflected in our revised manuscript, all key effects replicated in this within-subject design which in turn suggest that the findings of Experiment 1 are not the results of the lack of monetary incentives.

Of course, as mentioned also in our response to the minor point 5 of Reviewer 1, we cannot rule out that participants choice behavior might even look different when using larger incentives (e.g., 50€ vouchers) and/or other incentives structure outcomes (e.g., if every participants would have obtain real-word money in each trials instead of points) but it is beyond the scope of this study to investigate choice behavior as a function of stake/outcome size & structure.

Again, however, we can now see that design issues regarding outcome may be a critical factor in driving probabilistic decision-making. Thus, we have also decided to make readers aware of this issue early in our General Discussion on page 24. Although, it does not seem that participants choice behavior depends on the specific incentives used in the present study, it is possible that stronger (or weaker) incentives could influence how risk-seeking/averse participants behave in the loss and gain conditions, respectively.

The analysis of the post-experiment survey, while not central to the main argument, needs much greater rigor if it is to be included. For instance, on p. 27, the manuscript states that participants “rated described options as more likely to occur than experience ones and the size of this gap was largest for low-probability (20%) stimuli and tended toward inversion at high probability (80%).” The difficulty with this statement is that the stimuli were all binomial(n, p) lotteries. As first noted in Fox and Hadar (2006), this creates a confound due to the skewness of the binomial distribution, which is $(1 - 2p)/\sqrt{np(1 - p)}$. For $p < 1/2$, the skewness is positive: a small number of participants would be expected to observe more than np successes, and a large number would be expected to observe fewer. As p increases, this would diminish and tend toward inversion. A between-participants analysis is clearly inappropriate.

To correct for this, I suggest the authors conduct a within-participants analysis. For each participant, take the sample that the participant drew on the experience options. Compare the fraction of successes to the corresponding described probability. In the cases in which the sample exceeds the described probability a of 20%, did the participants still treat the described probability as more likely? Then do the same analysis for described probabilities of 50% and 80% and the corresponding samples on an individual level for the description options. Is there a systematic variation with the success probability within participants?

Thank you for this thoughtful comment. We appreciate your concern and we agree. Thus, we have implemented the suggested analysis (detailed on page 22) and updated the corresponding figure, Figure 6.

To respond more directly to the question of whether participants still differently considered the experimental probabilities for descriptive and experience stimuli: as can be seen in the Figure 6B,C, the revised analyses still indicates that participants’ metacognitive beliefs regarding the probabilities were independent of their real choice behavior.

The discussion of risk versus uncertainty is at odds with conventional usage. Since Knight (1921), risk has been used to characterize settings with known probabilities, and uncertainty has generally been used to characterize settings without known probabilities. It is strange to claim that risk refers to “situations where outcomes are deterministic” (what exactly is at risk then?).

We apologize for the confusion regarding our prior definitions of risk and uncertainty. We have modified the sentence on page 27 according to your suggestion:

“Risk characterizes settings with known probabilities and uncertainty describes settings without known probabilities (Knight, 1921).”

We appreciate that our initial wording was unclear because we referred to risky situations as those “where outcomes are deterministic and the decision maker has all relevant

information...”. We can see now that the word deterministic in this phrase could be confusing (note however that the gist of your comment regarding the role of information distinguishing a situation as risky or uncertain was reflected in our initial wording).

Similarly, uncertainty does not conventionally have anything to do with learning, inference, or repeated experience. A decision maker can face a situation with basis for assigning any definite probability to any outcome (a point raised in Kolmogorov, 1983). Repetition has nothing to do with this.

We think there may be a misunderstanding – we are not thinking of uncertainty in the strictly probabilistic sense but rather from a behavioral modelling/learning perspective (now emphasized on page 27), which places a large emphasis on the relationship between uncertainty (reduction) and learning.

This is precisely because “a decision maker can face a situation with basis for assigning any definite probability to any outcome” that we have outlined our model in terms of a distribution. The essential question being: what is the probability that a given probability is the true underlying one? Phrased another way, the underlying probability associated with an option is a hidden variable that a subject can only approximate through repeated. Any decision maker learning to approximate the underlying probability will require repeated exposure to narrow down the possibility space, thereby reducing uncertainty - a process one might reasonably term inference. In line with this perspective, we note that Reviewer #1 (major comment 6) explicitly asked to consider making the link between uncertainty and learning more explicit in terms of information-seeking behavior.

Some of the other references to the literature need more careful statements. For instance, the manuscript reads as if Kahneman and Tversky (1979) study probabilities that are inferred through experience (their focus is elsewhere, specifically on questioning the appropriateness of expected utility theory and drawing attention to the idea probability weighting function). A better early reference is the study by Davidson et al. (1957), in which participants are given a specially manufactured die with nonsense syllables on each face. The participants are allowed to practice with the die, in order to give them the experience of equally probable outcomes.

We are sorry that we have been not sufficiently clear in our wording when referring to previous studies. As we hope becomes clear from our two previous responses, we have slightly revised statements in the context of the risk/uncertainty literature to avoid any confusion. Furthermore, we have thoroughly checked the remaining manuscript with regard to the cited message to make sure we correctly convey the findings and claims of previous studies.

To respond more directly to the concern about the Kahnemann and Tversky (1979) study: Yes, the reviewer is correct that Kahnemann and Tversky (1979) did not directly study

probabilities that are inferred through experience. When we cited this study after a sentence, we had intended to refer to the part of the sentence that relies on the basic claim of this paper (in terms of prospect theory) that probability perception depends on the probability in question. However, we can see that this confusing. In our revision, we have decided to drop the reference and modified the corresponding sentences to not depend on the reference.

A better early reference is the study by Davidson et al. (1957), in which participants are given a specially manufactured die with nonsense syllables on each face. The participants are allowed to practice with the die, in order to give them the experience of equally probable outcomes.

Thank you for pointing this study out to us. Unfortunately, we were unable to obtain a copy of the book and it was not in the UC Berkeley or Tuebingen University catalogue. While we appreciate you making us aware of this reference we would prefer not to cite it since we have not been able to obtain and read it for ourselves and continue to cite only Hertwig et al.'s (2004) study.

The manuscript also states that most prior work gives a choice between an experience option and a described certain option. This is not the case in Hertwig et al. (2004), which the manuscript cites.

Yes, the reviewer is correct that Hertwig et al. (2004) did not use a condition where subjects chose between an experience option and a described certain (100%) option. However, in this (as in most previous previous studies), decision from experience and decision from descriptions were studied in different trials (not in equiobaly trials as we did in the present study). For example, in the Hertwig et al. Study there was a description and an experience group. In our revision, we have rephrased the text and made this distinction more clear whenever possible.

As a final suggestion, I recommend that the authors try to state the research question more explicitly. The current writing discusses various comparisons, but does not state in a straightforward way what the study aims to test and what the broader significance of the question is.

Yes, this is a good suggestion and a similar point was also raised by Reviewer 1 (major point 5, please see also our response there). Initially, we set out to test whether the classic DE gap (*vis a vis* Hertwig et al and Abdellouai et al) would be directly observable in an equiprobable choice paradigm without any sort of intervening steps (e.g. computing a certainty equivalent). Our question was whether the inverse-S probability weighting predicted by prospect theory and its modulation by choice type (description, experience) would generalize to a straightforward choice task. We have implemented several changes throughout our introduction (highlighted in the manuscript) to make our experimental goals and potential contribution more clear.

The discussion of response times comes across as an aside, and the reasons for excluding fast response times (which might include participants who have a solid understanding of the task) and slow ones (which may include those who are learning) needs more justification and a better connection to the main research question. I understand that fast and slow respondents may be thought of as a source of noise due to lack of engagement, but it would be helpful to have some discussion and perhaps some reassurance that including the entire sample does not drastically affect the significance or direction of the findings.

The reviewer is correct that we excluded trials with response times (RTs) less than 100 ms or greater than 3000 ms as RT outliers from any analyses as some trials as RT outlier trials for all analyses. In general, it is a standard procedure in RT-based experiments to exclude RT outlier trials to obtain a more precise measurements of the variables of interest (i.e., very fast or slow responses are thought to primarily reflect noise instead of being the result of a deliberative choice process).

Because we were not aware of any previous experimental procedure which would have provided a priori guidelines, the specific cut-offs for Experiment 1 were chosen based on visual inspection of the RT distribution as we have now specified in our result section. For Experiment 2, we had planned to use the same RT cutoffs but we also ensured that that these are appropriate by checking again the RT distribution.

We should also note that RTs <100ms would be faster than express-saccades, a form of rapid eye movement that is the fastest-known motor function of humans. It is not physiologically possible for RTs <100ms to be the result of a deliberative choice.

In general, it should be also emphasized these cut-offs were rather conservatively set and resulted in less than 4% of trials to be excluded in both experiments, respectively. Nevertheless, we have also checked whether the reported results are stable when using different outlier criteria (i.e., no RT cutoffs, RT cutoffs of 200 ms and 2000 ms, RT cutoffs of 50 ms and 4000 ms). These analyses revealed no qualitative differences in the result pattern (likely because so few trials were excluded to begin with).

We have made it clear that we decided the RT cutoffs based on visual inspection of the data and highlighted the very small proportion of trials which were defined as outliers in each experiment in the revised Methods section (see page 10 of revised manuscript).